# Composite Polylactide/Polycaprolactone Foams with Hierarchical Porous Structure for Pre-Vascularized Tissue Engineering

**DOI:** 10.3390/ijms26072974

**Published:** 2025-03-25

**Authors:** Jana Musílková, Miloš Beran, Antonín Sedlář, Petr Slepička, Martin Bartoš, Zdeňka Kolská, Šárka Havlíčková, Jitka Luňáčková, Lucie Svobodová, Martin Froněk, Martin Molitor, Hynek Chlup, Lucie Bačáková

**Affiliations:** 1Institute of Physiology of the Czech Academy of Sciences, Videnska 1083, 142 00 Prague, Czech Republic; antonin.sedlar@fgu.cas.cz (A.S.); lucie.svobodova@fgu.cas.cz (L.S.); 2Czech Agrifood Research Center, Drnovská 507/73, 161 01 Prague, Czech Republic; milos.beran@carc.cz (M.B.); martin.fronek@carc.cz (M.F.); 3Department of Solid State Engineering, University of Chemistry and Technology in Prague, Technicka 5, 166 28 Prague, Czech Republic; petr.slepicka@vscht.cz (P.S.);; 4Institute of Dental Medicine, First Faculty of Medicine, Charles University and General University Hospital in Prague, Kateřinská 32, 128 01 Prague, Czech Republic; martin.bartos@lf1.cuni.cz (M.B.); jitka.lunackova@lf1.cuni.cz (J.L.); 5Institute of Anatomy, First Faculty of Medicine, Charles University, U Nemocnice 3, 128 00 Prague, Czech Republic; 6Centre for Nanomaterials and Biotechnology, Faculty of Science, J. E. Purkyně University in Ustí nad Labem, Pasteurova 15, 400 96 Usti nad Labem, Czech Republic; zdenka.kolska@ujep.cz; 7Department of Plastic Surgery, First Faculty of Medicine, Charles University and Na Bulovce Hospital, Budinova 67/2, 180 81 Prague, Czech Republic; martinmolitor1@gmail.com; 8Department of Mechanics, Biomechanics and Mechatronics, Faculty of Mechanical Engineering, Czech Technical University in Prague, Technicka 4, 160 00 Prague, Czech Republic; hynek.chlup@cvut.cz

**Keywords:** degradable polyesters, three-dimensional scaffolds, macroporosity, nanoporosity, mineralization, compression stress and strain, mesenchymal stem cells, endothelial cells, dynamic cultivation, pre-vascularization

## Abstract

Modern tissue engineering requires not only degradable materials promoting cell growth and differentiation, but also vascularization of the engineered tissue. Porous polylactide/polycaprolactone (PLA/PCL, ratio 3/5) foam scaffolds were prepared by a combined porogen leaching and freeze-drying technique using NaCl (crystal size 250–500 µm) and a water-soluble cellulose derivative (Klucel^TM^ E; 10–100% *w*/*w* relative to the total PLA/PCL concentration) as porogens. Scanning electron microscopy, micro-CT, and Brunauer–Emmett–Teller analysis showed that all scaffolds contained a trimodal range of pore sizes, i.e., macropores (average diameter 298–539 μm), micropores (100 nm to 10 μm), and nanopores (mostly around 3.0 nm). All scaffolds had an open porosity of about 90%, and the pores were interconnected. The size of the macropores and the nanoporosity were higher in the scaffolds prepared with Klucel. Nanoporosity increased water uptake by the scaffolds, while macroporosity promoted cell ingrowth, which was most evident in scaffolds prepared with 25% Klucel. Human adipose-derived stem cells co-cultured with endothelial cells formed pre-vascular structures in the scaffolds, which was further enhanced in a dynamic cell culture system. The scaffolds are promising for the engineering of pre-vascularized soft tissues (relatively pliable 10% Klucel scaffolds) and hard tissues (mechanically stronger 25% and 50% Klucel scaffolds).

## 1. Introduction

Tissue engineering is an interdisciplinary field that combines functional cells, isolated from the patient’s organism and expanded in vitro, a biodegradable biomaterial, and appropriate biochemical and mechanical signals to promote the regeneration of diseased or injured tissue. A highly porous scaffold plays a critical role in positioning cells and guiding their growth and tissue regeneration in three dimensions.

One of the basic requirements for the long-term functionality of tissue substitutes is their ability to integrate with the surrounding tissue and to establish functional contact with the environment, in particular with an adequate blood supply to ensure the proper exchange of nutrients and gases with the environment, which is essential for cell survival. The creation of pre-vascularized tissue substitutes, in which the vascular network would be connected to blood vessels in the surrounding tissue, remains a challenge even in advanced tissue engineering [1,2,3,4]. Attempts to create vascularized tissues have focused on bioprinting channels in the engineered tissue, which then attract microvessel ingrowth from the surrounding tissue after implantation of the construct in vivo [5], or can be seeded with endothelial cells to mimic blood vessels under in vitro conditions [6]. Another promising approach to vascularize the engineered tissue is the direct incorporation of endothelial cells into a three-dimensional (3D) matrix. In our previous studies, and those of others, endothelial cells incorporated into collagen- or fibrin-based hydrogels spontaneously formed tubular capillary-like structures, especially when they came into contact with adipose-derived stem cells (ASCs), which were attached to these tubular structures like pericytes (for a review, see [7]). It is believed that the pre-capillaries in the tissue-engineered constructs, at least some of them, can spontaneously connect to the capillaries of the host tissues through a phenomenon called inosculation [2,8].

Vascularization of engineered tissues has typically been achieved using tissues based on natural polymers such as collagen or fibrin. However, a wide range of better-defined, degradable synthetic polymers are also available as scaffolds for tissue engineering (for a review, see [9]). Most of these polymers belong to the polyester family, which includes polylactides, polyglycolides, and polylactones. Polylactic acid (PLA) is more hydrophobic than polyglycolic acid (PGA) and more resistant to hydrolytic attack than PGA. For most applications, the L-isomer of lactic acid is chosen because it is preferentially metabolized in the body. Degradation of PLA, PGA, and PLA/PGA copolymers generally involves random hydrolysis of their ester bonds. PLA degrades to form lactic acid, which is normally present in the body. This acid then enters the tricarboxylic acid cycle and is excreted as water and carbon dioxide. PGA is also degraded by certain enzymes, especially those with esterase activity. The degradation rate of all these polymers can be further modulated by factors such as configurational structure, copolymer ratio, crystallinity, molecular weight, morphology, stresses, amount of residual monomer, porosity, and site of implantation. Polycaprolactone (PCL) is the most widely studied polylactone. PCL is a semi-crystalline polymer with a glass transition temperature of approximately −60 °C. The polymer has a low melting temperature (59 to 64 °C) and is compatible with a number of other polymers. PCL degrades at a much slower rate than PLA and is a useful base polymer for the development of long-term implantable drug delivery systems. Copolymers of ε-caprolactone with DL-lactide have been synthesized to produce materials with faster degradation rates (e.g., a commercial suture MONOCRYL, Ethicon). PLA, poly(lactic-*co*-glycolic acid) (PLGA) copolymers, PGA, and PCL are among the few biodegradable polymers with Food and Drug Administration (FDA) approval for human clinical use (for a review, see [9,10]).

For tissue engineering, synthetic polymers must be processed into porous 3D scaffolds. Strategies for fabricating 3D scaffolds include electrospinning, centrifugal spinning, molecular self-assembly, porogen (particulate) leaching, gas foaming, microfluidics, rapid prototyping including 3D printing, and phase separation including freeze-drying (for a review, see [11,12,13]). Each of these methods has advantages and disadvantages. For example, electrospun nanofibrous meshes mimic the architecture of the natural extracellular matrix (ECM) and have a high surface-to-volume ratio for binding cells and various bioactive factors, but are usually produced in the form of planar sheets rather than true bulky 3D scaffolds [14]. Centrifugal spinning should be used to produce bulky fibrous 3D scaffolds, but the pores in these scaffolds are usually too small for cell ingrowth into the material [15]. Gas foaming is a relatively simple technique that does not require the use of solvents, but scaffolds produced by this process often have poorly interconnected pores of variable size and a non-porous outer surface, making them less suitable for tissue engineering [12,13]. Higher interconnectivity of the pores and more precise control of their sizes can be achieved by freeze-drying, which was successfully used in our previous study on collagen-based scaffolds for bone tissue engineering [16], and also by porogen leaching, i.e., leaching of premixed water-soluble substances such as salt particles, poly(vinyl alcohol) (PVA), and poly(ethylene oxide) (PEO) [17]. In our previous studies, salt particles (sodium citrate or NaCl) were used to produce groups of PLGA scaffolds with different and relatively well-defined pore diameters ranging from 40 to 600 µm [10].

The optimal pore size for the ingrowth of different cell types has been extensively debated in the scientific literature. Globally, most studies agree on a pore size between 100 and 700 µm [18]. Pore sizes that are too large are not conducive to cell habitation, and pore sizes that are too small are not sufficient for cell migration and proliferation [19]. In our previous study on PLGA scaffolds, pores as small as 40 µm in diameter were bridged by human osteoblast-like MG 63 cells on the surface of the scaffold without these cells penetrating into the interior of the scaffolds [10]. A smaller pore size (<100 μm) promoted the formation of non-mineralized osteoid or fibrous tissue [2,20] and endochondral ossification [19]. The hypoxic state associated with these pores may also stimulate endothelial cell migration [21]. Medium pore sizes (approximately 100–400 μm) promoted chondrogenic cell differentiation and cartilage regeneration [22,23], and large pore sizes (400–600 μm) were suitable for the formation of vascularized bone tissue [24,25]. In our previous study on PLGA scaffolds, pore sizes of 400–600 μm were more suitable for colonization with human osteoblast-like cells than scaffolds with pore sizes of 180–200 μm and 250–320 μm [10].

However, bone and other tissues typically have hierarchically organized structures, ranging from macro- to micro- and nanostructures, and therefore require similarly organized scaffolds. Larger pores in scaffolds can provide space for cell attachment and growth, while smaller pores can facilitate the diffusion of nutrients, metabolites, and oxygen [19,26]. The internal structure of pore walls is also important for cell behavior. Nanoscale irregularities on these walls, such as nanocrystals, promote the adsorption of proteins that mediate cell adhesion and subsequent growth and differentiation [19]. Another important factor is the porosity of the scaffolds, i.e., the percentage of the total scaffold volume occupied by pores. For example, cancellous bone, which is characterized by a network of trabeculae, has a porosity of 40–95%, whereas denser cortical bone, which consists of osteons (i.e., longitudinally oriented cylindrical elements with concentric lamellae), has a porosity of only 5–25% [19]. At the same time, favorable high porosity and mechanical strength of scaffolds tend to oppose each other [26].

The aim of our study was to develop resorbable polymeric foams with a hierarchically organized porous interconnected structure prepared by combined freeze-drying and porogen leaching techniques from solutions of PLA and PCL in 1,4-dioxane. A blend of these two polymers was chosen to enhance the advantages and reduce the disadvantages of each individual polymer. PLA is more hydrophilic than PCL, making it more suitable for cell adhesion and growth [27], but it has a glass transition temperature above room temperature, making it hard and brittle. PCL, however, has a low glass transition temperature, making it tough but with a modulus an order of magnitude lower than that of PLA [28]. PCL is generally a more robust, hydrophobic, and crystalline polymer with slower degradation kinetics than PLA. PCL is often used as a toughening agent for PLA [28,29]. Blends of PCL and PLA can also be used to improve the ductility and biodegradability of PLA [29].

Based on this knowledge and our preliminary experiments, we chose a PLA/PCL ratio of 3/5 to achieve optimal mechanical properties, increase hydrophilicity, and ensure a sufficient biodegradation rate of the scaffolds. The foams developed in this study were designed for vascularized soft-tissue engineering, such as adipose tissue for use in reconstructive breast surgery, and, after mineralization, in simulated body fluid (SBF), potentially for bone tissue engineering. To verify their potential for these applications, the scaffolds were seeded with ASCs, which are capable of both adipogenic and osteogenic cell differentiation (for review, see [7]). To develop experimental pre-vascularization of the scaffolds, ASCs were co-cultured with human umbilical vein endothelial cells (HUVECs), and the process was further facilitated by a dynamic cell culture system providing fluid shear stress and perfusion of the scaffolds with culture medium.

## 2. Results and Discussions

### 2.1. Scanning Electron Microscopy (SEM) Morphology of the Scaffolds

Figure 1 shows the gross morphology of the prepared PLA/PCL foams (ratio 3/5 *w*/*w*; total polymer concentration in dioxane 4% *w*/*w*; Figure 1A), an SEM image of the microstructure of the mineralized foam scaffold (Figure 1B), and a detail of a pore formed by NaCl crystal leaching (Figure 1C). The prepared foams have a fairly regular structure with a bimodal micro–macro pore size distribution. Freeze-drying of the dioxane solution of PLA/PCL resulted in foams with micropores a few micrometers in size. The hierarchical structure with bimodal pore size distribution was obtained by adding NaCl porogen with defined crystal size distributions to the solutions prior to freeze-drying and leaching the porogen crystals from the freeze-dried foams with demineralized water. The resulting pores had an oval cross-section with a length of approximately 100 µm, and a width in the range of 25 to 45 µm (Figure 1B,C). Achieving oval pore lengths greater than 100 µm was difficult due to the shrinkage of the pores after washing out the porogen. The foams had an axially oriented regular internal structure (Figure 1B,C), which was particularly noticeable after diluting the total concentration of PLA and PCL polymers from 4% (*w*/*w*) to 2% (Figure 2).

In order to loosen the internal structure of the foams and limit the shrinkage of the foams, we decided to try the addition of Klucel hydroxypropyl cellulose, which we suspended in the dioxane solutions of PLA and PCL along with the NaCl porogen prior to lyophilization. Klucel hydroxypropyl cellulose is a non-ionic, water-soluble cellulose ether with a versatile combination of properties. It combines solubility in aqueous and polar organic solvents, thermoplasticity, and surface activity with the thickening and stabilizing properties of other water-soluble cellulose-based polymers. It can be used as a rheology modifier and has also been used in various peroral and transdermal drug delivery systems [30,31]. The addition of Klucel hydroxypropyl cellulose (from 10 to 100% *w*/*w* of the original dry matter of PLA and PCL polymers) changed the almost regular internal structure of the foams to a more chaotic one (Figure 3). We did not observe a consistent effect of Klucel hydroxypropyl cellulose on pore size increase at 10 and 25% *w*/*w* weight ratios, but there is an obvious pore size-increasing effect at 50% and 100% *w*/*w* hydroxypropyl cellulose. The pore size increased up to about 200 µm.

The addition of a water-soluble cellulose derivative in the range of 25% to 50% weight ratios loosened the internal structure of the foams and reduced the shrinkage of the foam structure after washing out the porogen. At the 100% Klucel weight ratio, excessive structural degradation, deterioration of mechanical cohesion, and violation of the outer contours of the scaffold had already occurred (Figure 3).

We did not observe any crystal formation or mineral deposition on the surfaces of the foams after the biomimetic mineralization (Figure 4), which could be explained by the hydrophobicity of the PLA/PCL blends. The hydrophobicity of 3D PCL scaffolds significantly limited the penetration of aqueous media, including SBF, into the 3D scaffolds [32]. In PLGA blended with varying amounts of Pluronic^®^F-108, the mineralization of these scaffolds in SBF increased with the surface hydrophilicity of the material [33]. The mineralization of a polymeric scaffold (e.g., PCL) can be improved by plasma treatment, which increases the surface wettability of the material [34]. Conversely, the mineralization of the material further increases its surface hydrophilicity [32].

Figure 4 also shows micropores several micrometers in diameter and a relatively rough surface of the scaffolds. Surface roughness plays an important role in cell adhesion, growth, and differentiation. Macroscale surface roughness, i.e., irregularities ranging from at least 100 μm to millimeters or larger, is usually favorable for the integration of the implant with the surrounding tissue, i.e., mechanical anchoring of the implant and the ingrowth of the surrounding tissue into the implant. Nanoscale surface roughness, i.e., irregularities equal to or less than 100 nm, is also beneficial, because it mimics the nanoarchitecture of the native ECM and promotes the adsorption of cell adhesion-mediating proteins, such as vitronectin and fibronectin (present, e.g., in the serum supplement in the culture media), in a suitable, nearly physiological geometric conformation [35] that provides cell adhesion receptors with access to specific sites in these proteins, such as amino acid sequences like arginin-glycin-aspartic acid (RGD), that serve as ligands for the adhesion receptors [36]. From this point of view, surface nanoroughness can be considered to act synergetically with the moderate hydrophilicity of the material surface, which also promotes the adsorption of cell adhesion-mediating molecules in bioactive physiological conformations. However, the microscale surface roughness (irregularities 1–100 μm) can be controversial. This roughness has been reported to promote osteogenic cell differentiation, but it may hinder cell adhesion, spreading, and subsequent proliferation, and thus the formation of a sufficient bone mass necessary for firm integration of the implant [37] (for a review, see [38]).

### 2.2. Micro-CT Analysis of the Scaffolds

Micro-CT was used as a 3D imaging technique to provide further information about the microstructure of the scaffolds. For ease of understanding, let us first define the important terms: Total Volume = volume of the analyzed volume of interest, Scaffold Volume = volume of the scaffold material (without porosity), Percent Object Volume = scaffold volume/total volume, Porosity = total porosity of the scaffold (= open porosity + closed porosity), Open Porosity = volume of porosity communicating with the external space, and Closed Porosity = volume of porosity within the scaffold material without communication with the external space.

The results of the micro-CT analysis are presented in Figure 5, which also shows the statistical significance of the differences obtained in the studied parameters between the experimental group of samples. The percent object volume (Figure 5A) ranged from 6.8% to 11.1% in the following increasing order: control material without Klucel (Ctrl), material with 10% of Klucel (1), material with 25% of Klucel (2), and material with 50% of Klucel (3). This means that the control sample contains the least amount of material and is therefore the most porous. Consistent with this, the 3D analysis showed that the porosity of the scaffolds was highest in the scaffolds prepared without Klucel and decreased with increasing amounts of Klucel (Table 1).

*The object surface area / material volume ratio*, i.e., the ratio of the total scaffold surface area to the volume of the scaffold material (Figure 5B), ranged from 230.3 mm^−1^ to 179.4 mm^−1^ in decreasing order Ctrl, 2, 1, 3, with the values in Ctrl being significantly higher than those in samples 1, 2, and 3, i.e., in samples with 10, 25, and 50% of Klucel. However, the *object surface density*, i.e., the ratio of the scaffold surface area to the volume of interest, i.e., the total evaluated volume of the scaffolds (Figure 5C), ranged from 14.6 mm^−1^ to 19.9 mm^−1^ in increasing order 1, Ctrl, 2, 3, with the value in sample 3 being significantly higher than the value in sample 1. This parameter describes the area available for further modification or interaction with cells during material testing, cell seeding in vitro, or after insertion into tissue in vivo, and this area increases with increasing amount of Klucel used during the scaffold preparation.

The structure thickness, i.e., the average thickness of the walls of the structure (Figure 5D), ranged from 14.2 µm to 15.5 µm in increasing order 2, 1, Ctrl, 3. The differences in the mean values are small, although statistically significant. The walls of sample 3, i.e., the 50% Klucel sample, are significantly thicker than those of samples 1 and 2, i.e., the 10% and 25% Klucel samples. However, these results must be viewed in the context of the limitations of micro-CT imaging and its spatial resolution. In the case of thin pore walls consisting of a few pixels/voxel, the values will be a multiple of the pixel/voxel size value (5 µm in our study). Since the average is calculated, the result can be any number, but the local thickness is limited to the values of 5 µm, 10 µm, etc. These results are consistent with 2D and 3D visualizations, which show a fairly homogeneous structure of thin pore walls.

The structure separation, i.e., pore size (Figure 5E), ranged from 298 µm to 539 µm in increasing order Ctrl, 1, 3, 2, i.e., the pore size is larger in scaffolds prepared with Klucel than in those prepared without Klucel. This parameter is based on a 3D analysis using sphere fitting and is therefore independent of orientation, in contrast to 2D-based evaluation (e.g., SEM). However, both SEM and micro-CT results are in agreement that the presence of Klucel during scaffold preparation can increase the pore size of these scaffolds.

Thus, the pore size of our scaffolds, especially those prepared with Klucel, which averaged from 483 µm to 539 µm, can be considered suitable for bone tissue engineering. In our previous study and in studies by other authors, pores in a similar range, i.e., between 400 and 600 µm, appeared to be optimal for this purpose. These pores can accommodate the Haversian system of concentric lamellae in the cortical (i.e., compact) bone together with vascularization, and are also suitable for trabecular (i.e., spongy or cancellous) bone reconstruction [10,24,25]. However, too large a pore diameter may be counterproductive. It can result in reduced mechanical stability and internal surface area of the scaffolds, limited cell adhesion, insufficient cell-to-cell contacts, reduced cell proliferation activity, and mineralization, and, consequently, limited bone formation [19]. However, the definition of pores that are too large for bone formation varies between materials—for example, in 3D printed ceramic materials, such pores were over 1.5 mm in diameter, whereas in collagen-based scaffolds, such pores were only 500 μm in diameter [39,40].

However, we also evaluated the representation of pore sizes below 298 microns, namely, those between 10 and 200 µm, as they are also promising in terms of tissue healing, especially for soft tissues [2,20], cartilage [22,23], and endochondral ossification [19,41]. A minimum pore size of at least 100–200 µm is required for bone tissue engineering, because a normal Haversian system of concentric lamellae reaches at least this diameter (for a review, see [41]).

The results of the pore distribution in the 10–200 µm range are presented in Figure 6 and show differences between the groups of scaffolds studied. These pores are located in the walls between the largest pores formed by washing the NaCl crystals. It is evident that up to a pore size of about 60 µm, the control scaffolds have a higher percentage of pores than the scaffolds prepared with Klucel, especially with 10% Klucel. However, in the pore size range from approximately 70 µm to 100 µm, the highest percentage of pores was achieved in the scaffolds prepared with 50% Klucel, and the pore size above 100 µm was most prevalent in the scaffolds prepared with 10% Klucel. The lowest representation of pores between 10 and 200 µm was found in scaffolds prepared with 25% Klucel, but, as shown by the structure separation parameter, these scaffolds mainly contained pores larger than 500 µm (Figure 5E).

Finally, micro-CT analysis showed that the prepared foams had a very high total porosity of about 90% (Table 1). Porosity refers to the ratio of pore volume to total material volume, which is a material-independent morphological property [41]. In the case of our scaffolds, it was almost exclusively open porosity, i.e., the percentage of the volume inside the specimen that is connected to the boundaries of the volume of interest, i.e., to the external space, via porous spaces. In contrast, closed porosity is the volume of pores that are completely surrounded by the scaffold material. Based on micro-CT imaging, all our specimens are highly porous with interconnected pores, which is supported by the finding of a very low closed porosity below 0.01%. However, we should again consider the spatial resolution as a limitation of micro-CT imaging. There may be pore walls below the spatial resolution that remain undetected.

The addition of Klucel hydroxypropyl cellulose during the foam preparation slightly reduced the total porosity at weight ratios above 25%. However, the samples with Klucel had a higher content of larger pores than the control samples without Klucel, as shown in selected representative morphological images from micro-CT analysis (Figure 7).

Micro-CT images also show that the pores in the scaffolds are generally well interconnected. Achieving a perfectly interconnected porous structure with favorable surface properties and considerable mechanical strength, which is required to meet the demands of cell proliferation, migration, and differentiation, as well as nutrient and waste delivery systems, is a major challenge. At the same time, desirable high porosity and mechanical strength tend to be mutually exclusive. The fabrication of highly interconnected porous structures with adequate mechanical strength usually involves several methods, mainly thermally induced phase separation, particle leaching, solvent casting, injection molding with sacrificial leaching, or 3D bioprinting [26].

### 2.3. Brunauer–Emmett–Teller (BET) Analysis of the Scaffolds

While the SEM analysis shows the bimodal micro–macro pore size distribution, the BET analysis revealed the existence of a third level of porosity of the foam scaffolds, i.e., the nanoporosity in the range of approximately 1–25 nm in half-pore width, i.e., 2–50 nm in diameter, which is below the resolution of the SEM used in our study, with the maximum peak around 3 nm in diameter in this part of the pore size distribution spectrum (Figure 8). The use of Klucel hydroxypropyl cellulose in the foam production process significantly increased the nanoporosity of the foams. Table 2 shows a statistically significant increase in surface area and nanopore volume after the addition of Klucel hydroxypropyl cellulose. It is important to note that the BET analysis method used is only suitable for measuring pores smaller than 40 nm. The presence of larger pores did not affect the results obtained.

For better illustration and clarity, we have avoided using the current IUPAC pore size classification scheme [42] in the text. Instead, we have used the more illustrative traditional categories of nanoporosity (the pore sizes < 100 nm), microporosity (the pore sizes ranging from 100 nm to 10 μm), and macroporosity (the pore sizes ranging from 10 μm to 1 mm). All three identified levels (nanoporosity, microporosity, and macroporosity ranges) in the trimodal spectrum of pore size distribution can play a significant role in the colonization of the foam by cells. Macroporosity (pore sizes in the range of tens to hundreds of µm) allows cells to penetrate the scaffold structure and migrate throughout its volume. High macroporosity facilitates better cell infiltration, which is essential for homogeneous cell distribution within the scaffold and uniform tissue formation. Microporosity (pore sizes in the range of units to tens of µm) can improve the diffusion of nutrients, oxygen, and waste products throughout the scaffold matrix. Adequate porosity ensures that cells located deep within the scaffold receive sufficient oxygen and nutrients for metabolic activities, thereby promoting cell viability and tissue growth. Microporosity can also promote the deposition of ECM proteins such as collagen, elastin, and glycosaminoglycans, which are critical for tissue development and functionality due to the increased surface area (for a review, see [18,43]).

The addition of Klucel hydroxypropyl cellulose had a significant effect on increasing the nanoporosity in the range of 2.4–6.0 nm (corresponding to the half-pore size of 12–30 Å) (Figure 8A,B; Table 2), resulting in a significant increase in the total water uptake of the foams (see Figure 9). This can be explained by capillary suction, which allows the foam to absorb water even against gravity. This is due to the intermolecular forces between the liquid and the surrounding solid surfaces. If the diameter of the tube is small enough, the combination of surface tension (caused by cohesion within the liquid) and adhesive forces between the liquid and the container wall will act to propel the liquid. Conversely, larger pores may not exhibit such strong capillary suction and may not retain water as effectively. The increase in nanopore volume in the original micro–macro-porous structure provides high capillary pressure, while the large pores provide high permeability of the foams. The effect of nanostructure on capillary performance has been described in [44]. These authors developed a model to illustrate the effect of both nanostructure and microstructure on the transport of working fluid in the biporous foam. The data indicate that the nanostructure generates high capillary pressure, while the microstructure provides low resistance pathways for rapid fluid transport. The size of the micropores and the ratio of micropores to nanopores were two important factors. The effects of different scaffold pore sizes have been reviewed by [18]. These authors provide an overview of processes that can be regulated by simply changing the pore size of the scaffold and how their targeted application could support tissue engineering.

The surface area of the samples increased dramatically from the control sample without Klucel (CTRL) to the samples modified with 10, 25, and 50% Klucel. The increase in surface area is caused by the increase in pore volume in the samples (Table 2, 3rd column). This increase can be seen in Figure 8, which shows a histogram of the pore volumes of individual samples. A detailed study of this histogram shows that the distribution of pore widths varies, especially for the half-pore widths of 10–50 Å (i.e., 1–5 nm). On one hand, the smallest pore volume is typical for CTRL and the highest for samples with 10% Klucel, while the pore volume continues to decrease for samples with 25 and 50% Klucel. On the other hand, the representation of pores with larger widths increased more in samples with 50% Klucel than in samples with 25 or 10% Klucel (Figure 8).

### 2.4. Water Uptake by the Scaffolds

The results of the determination of water uptake by the investigated foams (PLA/PCL 3/5 *w*/*w*; total concentration of polymers in dioxane 4% *w*/*w*) prepared without or with the NaCl porogen and different weight ratios of Klucel hydroxypropyl cellulose are shown in Figure 9. The addition of Klucel hydroxypropyl cellulose to the foams resulted in a statistically significant increase in water uptake. The increase was even more pronounced in foams prepared without the addition of the NaCl porogen, i.e., without macropores. However, in foams prepared with NaCl, the water uptake was generally higher (from 19.48 ± 0.15 to 22.04 ± 0.14 g/g) than in foams without the NaCl porogen (from 5.81 ± 0.10 to 11.75 ± 0.06 g/g; cf. Figure 9A,B). The increase in water uptake corresponded well with the increase in nanoporosity in the range of 12–30 Å half-pore width, as shown by BET analysis (Figure 8B and Table 2).

Both pore size and distribution affect water uptake. Foams with higher pore volumes, especially those with smaller and interconnected pores, can hold more water due to the increased surface area available for water retention. The connectivity of the pores within the foam structure also plays an important role. Foams with well-connected pores can facilitate the movement of water throughout the structure, allowing for more efficient water uptake. In contrast, isolated or poorly connected pores can limit water penetration and uptake. Overall, the pore size in foams has a direct effect on their ability to absorb water. Smaller, well-connected pores with high capillary suction and suitable surface properties generally result in greater water uptake capacity, whereas larger or poorly connected pores can limit water retention. The water-holding capacity of a material can also be increased by making it more hydrophilic, i.e., by applying hydrophilic Klucel or a natural mineral, represented, e.g., by NaCl.

### 2.5. Fourier-Transform Infrared Spectroscopy (FTIR) Characterization of the Scaffolds

Infrared (IR) spectroscopy was used as a method to identify the chemical structure of the scaffolds and the changes in the scaffold surface after the biomimetic mineralization.

In the FTIR spectra of the prepared foams, the characteristic bands of PCL predominate (2950–2860 cm^−1^, 1725 cm^−1^, 1240 cm^−1^, and 1190 cm^−1^), and the characteristic bands of PLA appear in the spectrum mainly in the region around 1080 cm^−1^. The FTIR spectra of the materials with and without NaCl porogen were very similar and virtually identical, indicating that the NaCl porogen was completely washed out of the material, leaving no residual spectrum.

After the biomimetic mineralization of the foams, we observed only very slight changes in the FTIR spectra of the prepared foams in the form of weak bands at 1214 cm^−1^ and 1137 cm^−1^, and an increase in the intensity ratio of the PLA bands, i.e., the PLA vibrations C=O at 1750 cm^−1^, C-H at 1460 cm^−1^, and C-O-C at 1090 cm^−1^ were more pronounced (Figure 10A).

IR spectroscopy was also used to identify possible residues of Klucel hydroxypropyl cellulose after thorough washing of the foams with demineralized water. On the one hand, the use of Klucel in the preparation of foams had no significant effect on the IR spectra of the prepared foams at any concentration. On the other hand, small changes in the IR spectra after the application of Klucel compared to the control samples without Klucel occured, and were similar in character to the changes caused by biomimetic mineralization. Again, there is an increase in the intensity of the PLA bands (Figure 10B). While PCL improves the mechanical properties of PLA/PCL blends [28,29], PLA as a more hydrophilic polymer increases the wettability of PLA/PCL blends [27], and this wettability is likely to be further improved by the application of hydrophilic Klucel [30,31] and the introduction of the mineral component [32]. The application of Klucel during the scaffold preparation, similar to the mineralization of the surface, probably altered the polymer surface properties (e.g., by some retention of Klucel on this surface), which may have contributed to the improved hydrophilicity of the scaffolds (Figure 9). The surface properties of the foam, including its wettability, can influence water uptake. It is known that hydrophilic surfaces tend to attract and retain water more readily, whereas hydrophobic surfaces can repel water, thus affecting how water is absorbed into the foam. The wettability of the material surface also affects the cell adhesion to the material surface. Cells spread, proliferate, and differentiate better on hydrophilic surfaces, which adsorb less protein than hydrophobic surfaces, but in a more physiological conformation that can be recognized by cell adhesion receptors, such as integrins (for a review, see [38]). The wettability of the scaffold surface is also related to its topological structure. For example, nanostructuring of the surface of the biomaterial increases its hydrophilicity [45].

### 2.6. X-Ray Diffraction (XRD) Analysis of the Scaffolds

Since FTIR did not completely rule out the possible retention of Klucel in the scaffolds, we proceeded to analysis by XRD. XRD analysis was performed for various weight ratios of Klucel hydroxypropyl cellulose incorporated into PLA/PCL foams. It is evident that all NaCl was completely washed out during the scaffold preparation. PLA and PCL show high crystallinity; for PLA, it is due to diffractions at 16.6°, 22.6°, and 50.5°, and corresponding crystalline diffractions for PCL are at 16.7°, 21.4°, 23.8°, 43.6°, and 44.7°. It is noteworthy that in the presence of Klucel hydroxypropyl cellulose, a characteristic rounded peak appears at the corresponding position, which was previously reported in a study by Basta et al. [46] focusing on hydroxypropyl cellulose-based liquid crystal materials, and is marked as a gray column area in Figure 11.

### 2.7. X-Ray Photoelectron Microscopy (XPS)

The elemental spectra of the samples studied are shown in Figure 12. It is evident that the incorporation of Klucel into the foams during their preparation increases the oxygen atomic concentration (approx. 31.3–34.1 at.%) compared to the control without Klucel (only 28.8 at.%). This is in good agreement with the chemical structure of the Klucel molecule. Interestingly, the oxygen atomic concentration did not increase proportionally with the concentration of Klucel used to prepare the scaffold, being highest in scaffolds prepared with 10% Klucel (34.1 at%) and lowest in scaffolds prepared with 25% Klucel (31.3 at.%), which suggests the lowest retention of Klucel in the latter scaffolds.

In addition, an insignificant amount of chlorine was detected in the sample prepared with 50% Klucel (0.19 at.%), which may indicate a trace concentration of NaCl in this sample. Thus, similarly to XRD, XPS analysis confirmed some retention of Klucel in the scaffolds. In the case of scaffolds prepared with 50% Klucel, some retention of NaCl porogen cannot be excluded, but only in trace amounts.

### 2.8. Mechanical Testing of the Scaffolds

The prepared PLA/PCL foams (ratio 3/5 *w*/*w*; total polymer concentration in dioxane 4% *w*/*w*) were also mechanically tested by compression in a Zwick/Roell biaxial test system designed for mechanical testing of soft tissues and polymers. Each specimen was compressed between two plates, and the initial thickness (H) and deformed thickness (h) were determined. Since mineralization is believed to improve the mechanical properties of polymeric scaffolds (for a review, see [47]), non-mineralized control samples (NM) were compared with mineralized samples without Klucel (0%) and with 10%, 50%, and 100% Klucel.

When the H thicknesses of the specimens were analyzed, some groups were found to be significantly different (Figure 13A). The thicknesses of the NM samples and those prepared with 10% and 100% of Klucel were similar, while the thickness of the samples prepared with 25% and 50% Klucel was about half that of the other groups. Interestingly, the mineralized samples without Klucel (0%) were slightly but significantly thinner than the unmineralized samples, which may be due to the fact that the collapse of the 0% samples after immersion in the SBF was not prevented by Klucel.

The area S of the samples where the force was applied also differed among the samples. This was due to the fact that when the experimental specimens were prepared for mechanical testing, it was difficult to produce similar specimens due to the structure of the foams. This is shown by the analysis of the specimen surfaces in Figure 13B. The foams of the 25% and 50% groups were able to produce specimens with smaller initial area (S), calculated as width (A) × height (B), than the others. The 25% and 50% Klucel groups again differed from the others. This conclusion indicates that stress, not force, must be compared.

The obtained experimental stress–strain (σ−ε) characteristics are shown in Figure 14A. The variance of the mechanical response of the samples of each group can be seen. Since PLA/PCL scaffolds represent a porous material made of organic substances, the scatter of the data (and thus the repeatability of the measurements) is acceptable. The σ−ε characteristics of the non-mineralized (NM) and mineralized (0%) control groups overlap. This is similar to the 10% and 100% groups. This can be seen in the calculated representative average characteristics with confidence intervals (Figure 14B). The σ−ε characteristics of the 25% and 50% Klucel groups differ from the NM and 0% groups in that they are steeper at higher values of the strain. This means that the samples are stiffer. The opposite is true for the 10% and 100% Klucel groups, which are below the NM and 0% groups. Thus, the 10% and 100% groups of samples are more pliable and less stiff (Figure 14B).

Similar conclusions can be drawn from Figure 15, which shows the strain (A) and stress (B) at maximum load. It can be seen that the 25% and 50% Klucel groups had the lowest strain values but the highest stress values at maximum load. The 10% and 100% Klucel groups had higher strain values compared to the NM and 0% groups (Figure 15A). However, the stress values were similar (see Figure 14 and Figure 15B).

When analyzing the strain energy values obtained, i.e., for smaller strain values up to 0.35, the above differences are even more pronounced. The 10% and 100% Klucel groups achieve similar strain energy values for all three selected strain states (strain 0.1, 0.25, 0.35), i.e., W_0.1_, W_0.25_, and W_0.35_. These values are much lower than for the 25% and 50% groups; see Figure 16. Up to strain 0.1, the mechanical response of the 25% and 50% groups is similar (Figure 16A). At higher strains, however, the mechanical properties of the 25% and 50% groups are different (Figure 16B,C). This can also be seen in Figure 14, where the σ−ε characteristics diverge, and in Figure 15 and Figure 16D, where statistical differences in ε_end_, σ_end_, and also in the compressive modulus E_end_, were found between the 25% and 50% groups of samples.

Taken together, the measurements of mechanical properties showed that there were three types of specimens. The first group is samples prepared without Klucel, either non-mineralized (NM) or mineralized (0%). These samples have relatively low stiffness, and, more importantly, their stiffness is similar, so it is evident that the mineralization in SBF did not have a significant effect on the mechanical properties of the PCL foams. The scaffolds prepared without Klucel, both mineralized and non-mineralized, had an elastic modulus *E* of about 1 MPa, which is close to the *E* of soft tissue, which ranges from 0.1 MPa to 1 MPa when tested in tension [48]. In tissue engineering, they could be used as scaffolds for soft tissue cells, e.g., for the reconstruction of adipose tissue, or as hemostatic foams.

The mechanical properties of the scaffolds were significantly influenced by the concentration of Klucel used in the preparation of the samples. Consequently, a second group of higher stiffness samples was created that included samples prepared with 25% and 50% Klucel. The deformation of these foams requires more load, i.e., more deformation energy. The application potential of these scaffolds could be in hard-tissue engineering, such as bone or cartilage reconstruction. However, it should be noted that the Young’s modulus (*E*) of bone tissue is in the range of units to tens of GPa, whereas our scaffolds prepared with 25% or 50% Klucel have an *E* of only about 2.5 to 3 MPa. Thus, they may be more suitable for filling bone defects (some improvement in mechanical properties can be expected after bone tissue has formed inside the scaffolds) than for load-bearing applications, such as bone-integrating parts of large joint replacements. These applications still require metallic materials, such as titanium and its alloys. Virtually the only polymer that could be used in place of metals in high-stress applications is PEEK, especially when reinforced with carbon fibers (for a review, see [49]).

Then there is a third type of samples with much lower stiffness. These include samples prepared with 10% or 100% of Klucel. Significantly lower loads are sufficient to deform these foams. Similarly to the scaffolds prepared without Klucel, they could be used for soft-tissue engineering. Representatives of all three mentioned groups, namely, scaffolds prepared with 0%, 10%, 25%, and 50% Klucel, were selected for further cell growth assays in static and dynamic cell culture systems.

### 2.9. Cell Growth on the Scaffolds

#### 2.9.1. Comparison of Non-Mineralized and Mineralized Scaffolds Prepared Without Klucel

In bone tissue engineering, mineralization is often believed to improve the mechanical properties of scaffolds made of synthetic and natural polymers (for a review, see [47]), which, however, was not demonstrated in our present study. In our previous study performed on electrospun PLA nanofibrous scaffolds, the addition of hydroxyapatite suppressed the creep behavior of the scaffolds in their dry state, but wetting of these samples promoted their creep behavior [14]. Biomimetic mineralization of porous collagen scaffolds in SBF in another study by our group [16] did not significantly change the mechanical stability of the scaffolds, i.e., their resistance to mechanical damage, but this resistance tended to be slightly higher in non-mineralized scaffolds. In a study by Tran et al. [47], prolonged mineralization of gelatin–alginate scaffolds in highly concentrated SBF even worsened the mechanical strength and structural integrity of these scaffolds.

Nevertheless, adequate mineralization of the scaffolds in SBF improved the cell adhesion, growth, viability, metabolic activity, and osteogenic differentiation, as well as the cell penetration of cells into the scaffolds [16,47]. As revealed by confocal microscopy in our present study (Figure 17), both non-mineralized and SBF-mineralized scaffolds were well colonized with cells. However, after 7 days of cultivation, the cells on the surface of the mineralized scaffolds were almost confluent, whereas on the non-mineralized scaffolds, there were still gaps between the cells and the cell distribution was less homogeneous. Calculation of the number of cells per cm^2^ of the Maximal Imaging Projection (MIP) of the surface of the tested materials showed that the mineralized scaffolds contained significantly more cells than the unmineralized scaffolds (Figure 18). These differences were observed both in standard growth medium and in differentiation medium supplemented with osteogenic factors, namely dexamethasone, β-glycerol phosphate, and ascorbic acid [7,50]. The higher cell number on mineralized scaffolds could be attributed to a biochemical stimulation of the cell growth due to the increased content of Ca, Mg, and P in the scaffolds. However, the higher level of cell confluence on mineralized scaffolds could also be due to the fact that the surface of these scaffolds appeared flatter than that of non-mineralized scaffolds (Figure 17).

#### 2.9.2. Comparison of Mineralized Scaffolds Prepared Without and with Klucel

Based on the previous finding that SBF-mineralized scaffolds supported cell adhesion and growth better than unmineralized scaffolds, only mineralized scaffolds were used for further cell experiments. Similarly to the previous experiments, the cells on mineralized scaffolds prepared with different amounts of Klucel were visualized by fluorescent staining of the filamentous actin (F-actin) cytoskeleton and cell nuclei, followed by confocal microscopy (Figure 19). The images taken from the top as a MIP showed that on day 6 of static cultivation, the cells on the scaffolds prepared with Klucel were better spread, i.e., spindle-shaped or polygonal, whereas the cells on the control scaffolds without Klucel were often rounded. Cells on the Klucel-treated scaffolds were also more homogeneously and more densely distributed, especially on scaffolds prepared with 25% Klucel. However, the calculation of the number of cells per cm^2^ of MIP of the surface of the tested materials revealed that these differences were not statistically significant, i.e., the cell population density on day 6 after seeding was comparable on all tested samples (Figure 20).

From day 6 to day 10, the cells were cultured on the scaffolds either in a conventional static system or in a dynamic system, i.e., a Stuart Mini Orbital Shaker SSM1, which provides orbital motion and exposes the cells on the scaffolds to the flow of culture medium. In the static system, the cell numbers increased significantly only on the control samples without Klucel, but not on the Klucel-treated materials, where the cell numbers remained similar to those found on day 6 and became significantly lower than those on the control samples without Klucel (Figure 19 and Figure 20). This result could be explained by the relatively high porosity and surface area of the non-Klucel scaffolds (Figure 5, Table 1), which provided sufficient space for cell attachment and growth. Consistent with this, the 3D side view showed that some cells on the scaffolds prepared without Klucel and also with 10% Klucel were able to penetrate into the interior of the scaffolds (Figure 19), whereas on the scaffolds prepared with 25% and 50% of Klucel, the cells were mainly concentrated in the superficial parts of the scaffolds. The regular parallel arrangement of pores in the scaffolds prepared without Klucel (Figure 1, Figure 2 and Figure 3) may also have played a favorable role in facilitating cell accommodation in these scaffolds, whereas the deformed pore structure in the Klucel-treated scaffolds likely hindered cell penetration into the scaffolds.

However, the cells on Klucel-treated scaffolds were generally better spread (i.e., more flattened) than on the scaffolds without Klucel (Figure 19 and Figure 21). This could be explained by an improved wettability of the Klucel-treated scaffolds, caused by a partial retention of a hydrophilic Klucel in the scaffolds, as indicated by XRD and also suggested by FTIR (Figure 10 and Figure 11). The moderate hydrophilicity of the material surface promotes the adsorption of cell adhesion-mediating molecules (vitronectin, fibronectin) in a bioactive physiological conformation, which further promotes the binding of specific amino acid sequences (e.g., RGD) in these molecules by cell adhesion receptors [36,38]. Another advantage of Klucel was its ability to modulate the mechanical properties of the scaffolds for different applications. As mentioned above, soft and more pliable scaffolds prepared without Klucel or with a low concentration of this compound (10%) seem to be suitable for soft-tissue engineering, while scaffolds prepared with 25 and 50% Klucel became stiffer and more mechanically resistant and thus more suitable for hard-tissue engineering.

The disadvantage of Klucel was that, in addition to deforming the porous structure, this compound remained trapped in the scaffolds, further hindering the penetration of cells into the interior of the scaffold. However, this obstacle could be overcome, at least in part, by growing cells on the scaffolds in a dynamic culture system.

Dynamic cultivation markedly improved cell spreading and F-actin cytoskeleton formation in the cells on all samples tested (Figure 21). Compared to day 6 after seeding, after an additional 4 days of dynamic cultivation, the cell number increased significantly not only in samples without Klucel, but also in samples with 25% of Klucel (Figure 20). In addition, the average cell number in the samples prepared with 25% Klucel was more than 2.5 times higher than in the corresponding samples grown for 10 days under static conditions, and the cell layer on their surface was almost confluent (Figure 21). Interestingly, XPS suggested the lowest retention of Klucel in these scaffolds (Figure 12). The average cell number also increased in scaffolds prepared with 50% of Klucel, but the differences between the values in the dynamic system (74,900 ± 8700 cells/cm^2^) and the static system on day 10 (37,800 ± 18,100 cells/cm^2^) or day 6 (37,800 ± 4100 cells/cm^2^) were not statistically significant (Figure 20). It is known that dynamic cultivation improves the supply of nutrients and oxygen to the cells, accelerates the removal of metabolic waste products, and allows mechanical stimulation of the cells, similar to physiological conditions in the body. Dynamic cultivation also improved cell ingrowth into the interior of the scaffolds, which was particularly evident in scaffolds prepared with 25% and 50% of Klucel. In these scaffolds, the cells penetrated to a depth of approximately 400 µm, whereas in the 10% Klucel samples, the cells appeared to follow a highly undulating surface relief with deep depressions (Figure 21).

Taken together, the highest colonization by ASCs was achieved in scaffolds prepared without Klucel, and with 25% and 50% Klucel. For scaffolds without Klucel, this result was achieved even in a static culture system. This type of scaffold may therefore be suitable for the construction of adipose tissue substitutes, since ASCs are de facto precursors of adipocytes, and their adipogenic differentiation is better stimulated in a static than in a dynamic culture system. Dynamic systems generating different types of mechanical stress, such as stretch and strain [51,52], vibration [53], fluid shear stress [54], or perfusion combined with compression [55] reduced adipogenic differentiation in favor of osteogenic differentiation. Only perfusion or compression alone did not affect the adipogenic differentiation of ASCs [55]. Adipose tissue substitutes can be used, for example, for breast reconstruction after tumor removal [56]. Current clinical practice for breast reconstruction is to use acellular silicone implants or to fill defects with autologous fat grafts. Both approaches have disadvantages—silicone implants are associated with the risk of local and systemic immune reactions, i.e., breast implant illness (BII, for a review, see [57]), while fat grafts are often at least partially resorbed, significantly reducing their beneficial effect [56]. These problems could be overcome by using resorbable scaffolds seeded with autologous cells and pre-vascularized. These scaffolds are then progressively replaced with the patient’s new regenerated tissue to permanently fill the defect and remove the artificial material. Even scaffolds mineralized in SBF could be used for this purpose, as our measurements showed no statistically significant changes in stiffness due to mineralization.

The flow of culture medium in the dynamic Stuart Mini Orbital Shaker SSM1 system tends to stimulate osteogenic differentiation, as we have shown in our recent study performed on human bone marrow mesenchymal stem cells (bmMSCs) cultured on silicalite-1 films deposited on Ti-6Al-4V substrates [58]. Scaffolds prepared with 25% and 50% Klucel, where the highest colonization by ASCs was achieved in the dynamic cultivation system, would be suitable for the construction of bone tissue substitutes, especially for filling bone defects, as mentioned above. Our previous studies have shown that ASCs are capable of osteogenic differentiation, although to a lesser extent than bmMSCs [50]. However, ASCs are more easily accessible through liposuction of subcutaneous adipose tissue, which is a less invasive and less painful method than bone marrow aspiration [59], and their osteogenic differentiation could be further enhanced by a combination of perfusion and compression stress [55].

#### 2.9.3. Cocultivation of ASCs with HUVECs on the Scaffolds

Whether we use our newly developed scaffolds for soft- or hard-tissue engineering, we need to vascularize the 3D cell–material construct, which is essential for the survival of the construct when it is implanted in vivo. In dynamic cell culture systems, the 3D cell–material construct is perfused by the cell culture medium. However, after implantation in vivo, this construct is nourished only by diffusion of nutrients and oxygen from the surrounding tissue, until the vasculature develops spontaneously within the construct, i.e., the blood vessels of the surrounding tissue invade the construct. Vascularization within a few days of in vivo implantation is critical for the survival and function of tissue-engineered grafts of clinically relevant size. It is one of the major limiting factors in their establishment for patient treatment and is a critical step in restoring cellular homeostasis [1,2,3,4]. A non-vascularized, diffusion-dependent construct often fails due to necrosis of the central part. Therefore, it is highly reasonable to pre-vascularize the construct by developing a vascular bed during engineering in dynamic cell culture systems. This vascular bed would then be connected to the blood circulation of the recipient of the implant, e.g., by spontaneous inosculation of the pre-capillaries in the implant and the capillaries of the surrounding tissue [2,8].

Our previous studies and those of others have shown that the presence of ASCs in cultures of human vascular endothelial cells (ECs) stimulates the ECs to form tubular capillary-like structures, to which the ASCs are externally attached as pericyte-like cells (for a review, see [7]). This phenomenon readily occurs in the setting of soft hydrogels, such as fibrin or collagen, and we intended to test whether similar pre-capillary formation was also possible within a synthetic polymeric porous scaffold, especially when assisted by fluid shear stress and scaffold perfusion provided by the Stuart Mini Orbital Shaker SSM1.

Our initial [1,2,3,4] experiments were performed on mineralized scaffolds without Klucel, seeded with ASCs and HUVECs at different ASC/HUVEC ratios of 2:1, 1:1, 1:2, and 1:5. Both cell types proliferated well in co-culture, and after 3 days in static culture, the endothelial cells formed elongated structures that could be considered the base of the vasculature within the cell–material construct. The most promising ASC/HUVEC ratio for the spontaneous formation of capillary-like structures within our scaffolds was 2:1, i.e., with a higher amount of ASCs (Figure 22).

Too high a ratio in favor of endothelial cells resulted in the formation of a confluent layer of endothelial cells on the surface of the scaffolds, rather than cell penetration into the interior of the scaffolds. Similarly, in a study by Bersini et al. [1], aimed at engineering vascularized bone-mimicking tissue in vitro, the use of bone marrow mesenchymal stem cells and HUVECs in a 1:1 ratio led to the formation of thin, highly interconnected microvessels, while a 10:1 ratio was characterized by large cell clusters in a hydrogel matrix. On the one hand, these clusters gradually remodeled into tiny capillaries, which ensured a higher oxygenation of the tissue construct [1]. On the other hand, when the amount of mesenchymal stem cells, e.g., ASCs in our study, is higher, some of the ASCs adjacent to the endothelial structures can serve as pericytes, while the other part in the interstitium can be differentiated towards a desired cell type, i.e., either adipocytes or osteoblasts. For this purpose, serum-free and hybrid media can be used, allowing the coupling of vasculogenesis with adipogenesis [60] or with osteogenesis [61]. Another possibility is to combine vasculogenic microtissues with adipogenic or osteogenic microtissues [4].

However, our pre-vascularized tissue-engineered cell–material construct needs further improvement. The penetration of cells into the scaffold, which subsequently positively affects the formation of elongated pre-vascular structures, can be enhanced by centrifuging the scaffold together with the cell suspension during cell seeding (Figure 23).

Another option is to enlarge the pores using Klucel and, in particular, to culture the cells under dynamic conditions provided by the Stuart Mini Orbital Shaker SSM1 (Figure 24). This system not only exposes the cells to fluid shear stress, which promotes phenotypic maturation of endothelial cells (for a review, see [7]), but also ensures, at least partially, the perfusion of the pores by the culture media, which promotes cell penetration into the scaffolds and the formation of pre-vascular structures. In all cases mentioned above, the pre-vascularization was significantly improved even when ASCs and HUVECs were seeded in a 1:1 ratio.

The formation of capillary-like structures in our PLA/PCL scaffolds could also be enhanced by combining these scaffolds with hydrogels, either natural (collagen, gelatin, fibrin) or synthetic, loaded with cell adhesion-mediating oligopeptides such as Arg-Gly-Asp (RGD), growth and angiogenic factors, such as vascular endothelial growth factor (VEGF), and other bioactive molecules (for a review, see [1,2,3,62]).

Another possible improvement in the construction of the vascularized adipose or bone tissue would be to use microvascular fragments (MVFs) isolated from subcutaneous adipose tissue instead of pure endothelial cells. In this study, commercially available HUVEC endothelial cells were used, but obtaining pure endothelial cells from patients can be a problem. ASCs are unable to fully differentiate into endothelial cells when stimulated by the usual physiological cues, such as the appropriate composition of the culture medium and exposure to laminar shear stress. This is mainly explained by the inability of ASCs to achieve the membrane polarization typical of endothelial cells, i.e., the functional specialization of the basal, apical, and lateral portions of the cytoplasmic membrane (for review, see [7]). Differentiated endothelial cells would have to be obtained from the patient’s subcutaneous veins, which would expose the patient to additional surgery. Therefore, it would be advantageous to enrich ASCs directly with the MVFs of subcutaneous adipose tissue obtained together with ASCs from the same lipoaspirate [63].

## 3. Materials and Methods

### 3.1. Preparation of the Scaffolds

#### 3.1.1. Preparation of the Soft Foams with Hierarchical Porous Structure from PLA, PCL, and Their Blends

Polylactide (PLA) Ingeo 4043D, Mn 160,000 g·mol^−1^, d-isomer content = 4.3% (NatureWorks, Plymouth, MN, USA), and polycaprolactone (PCL), average Mr 80,000 (Sigma-Aldrich, Merck Life Science Ltd., Prague, Czech Republic) and blends of PLA/PCL in the ratio of 3/5 were used to prepare the soft foams. Individual polymers were dissolved in 1,4-dioxane. The concentrations of the polymers or their blends were in the range of 1–5% (*w*/*w*). Two types of porogens were used, namely NaCl and a water-soluble cellulose derivative Klucel™ E hydroxyl propylcellulose (Ashland, OR, USA). Finely ground NaCl (crystal size 0.25–0.5 mm) was used either alone (1.3 g of NaCl per 1 g of PLA/PCL dioxane solution) or together with Klucel™ E, which was suspended together with NaCl in the polymer solutions at weight ratios of 10, 25, 50, and 100% *w*/*w* relative to the total concentration of PLA and PCL polymers. The reason for using Klucel was to further enlarge the pores and/or prevent them from collapsing. After freezing at −75 °C, the suspensions were freeze-dried in a Lyovac GT 2 freeze dryer (LH Leybold, Cologne, Germany). In the next step, the salt and Klucel were washed out of the material first by running deionized water for 1 day and then by rinsing the material five times with deionized water.

#### 3.1.2. Mineralization of the Prepared Foams

The prepared foam scaffolds were mineralized in simulated body fluid (SBF), which was prepared according to the protocol published by [64]. The SBF was sterilized in polycarbonate bottles in a Melatronic 15 N+ autoclave (Melag, Berlin, Germany). A quantity of 200 ml of the SBF solution was then used to immerse each gram of the freeze-dried foams. The foams were incubated in the SBF solution at 36.5 °C for 15 days with slow stirring. After 15 days, the solution was changed to a fresh one, and the foams were incubated for the next 15 days. Finally, the foams were repeatedly rinsed with an excess of demineralized water to remove residual inorganic ions. All chemicals were purchased from Sigma-Aldrich, Merck Life Science Ltd., Prague, Czech Republic).

#### 3.1.3. Sterilization of the Prepared Foams

Before the cell culture experiments, the samples of foams were sterilized by gamma-irradiation, using the basic sterilization dose (Microtron MT25, Nuclear Physics Institute of the Czech Academy of Sciences, Řež near Prague, Czech Republic).

### 3.2. Material Characterization Techniques

#### 3.2.1. Visualization of Scaffold Microstructure by Scanning Electron Microscopy (SEM)

SEM scans were obtained using a scanning electron microscope (FIB-SEM, LYRA3 GMU, Tescan, Brno, Czech Republic). The applied acceleration voltage was 10 kV. The investigated samples were covered with a 20 nm thick Pt conductive layer.

#### 3.2.2. Micro-CT

Micro-CT was used as a 3D imaging technique to provide information on the microstructure of the material. The specimens were scanned using the desktop micro-CT SkyScan 1272 (Bruker micro-CT, Kontich, Belgium) with the following scanning parameters: pixel size = 5 μm, source voltage 50 kV, source current 200 µA, no filter, rotation step  =  0.2°, frame averaging (5), specimen rotation of 180°, camera binning 2 × 2, scanning time approximately 1 h per specimen. Each specimen was scanned in a dry state while mounted on the specimen holder. The flat-field correction was updated prior to each scan. Bruker micro-CT software (Kontich, Belgium) was used for data processing. Data were reconstructed into cross-sectional images (NRecon software, version 2.0; Bruker micro-CT, Kontich, Belgium). Visualization was performed in 2D using DataViewer software (version 1.5; Bruker micro-CT, Kontich, Belgium) and in 3D using CTVox software (version 3.3; Bruker micro-CT, Kontich, Belgium). Data were processed to reduce image noise and binarized prior to 3D analysis to evaluate the following structural parameters:Total Volume = analyzed volume of interest.Scaffold Volume = volume of scaffold material (without porosity).Percent Object Volume = scaffold volume/total volume.Object Surface to Object Volume ratio.Object Surface Density (object surface to volume of interest ratio).Pore Size = mean pore size.Pore Size Distribution.Structure Thickness.Porosity = total porosity of the scaffold (= open porosity + closed porosity):○Open Porosity = volume of porosity communicating with the external space○Closed Porosity = volume of porosity within the scaffold material without communication with the external space.

The 3D analysis was performed using CTAn software (version 1.19; Bruker micro-CT, Kontich, Belgium). Pore size analysis was based on a sphere-fitting algorithm.

#### 3.2.3. Brunauer–Emmett–Teller (BET) Analysis

Specific surface area and pore volume were determined from nitrogen adsorption/desorption isotherms using a NOVA3200 instrument (Quantachrome Instruments, ANAMET, Prague, Czech Republic) with NovaWin software (version 11.03, Quantachrom Instruments, Boynton Beach, FL, USA). Samples were degassed for 24 h at room temperature. BET analysis was used to determine the total surface area, and the Barrett–Joyner–Halenda (BJH) model was used to determine pore volume and pore width distribution.

#### 3.2.4. Determination of the Water Uptake of the Scaffolds

Pieces of dry foams, prepared without and with the addition of NaCl porogen and without and with Klucel, weighing 0.3 to 0.5 g, were immersed in distilled water at ambient temperature. After two hours, they were removed from the water, blotted dry on filter paper to remove excess water, and weighed (W_w_). After freezing at −75 °C, the foams were freeze-dried in a Lyovac GT 2 freeze dryer (LH Leybold, Germany) and weighed (W_d_). Water uptake (WU) was calculated using the following formula:WU (%) = (W_w_ − W_d_)/W_d_ × 100(1)

#### 3.2.5. Fourier-Transform Infrared (FTIR) Analysis of Scaffold Surfaces

Changes in the composition of the scaffold surfaces after the addition of the Klucel hydroxypropyl cellulose and after biomimetic mineralization were evaluated by infrared spectroscopy analysis, performed using a Nicolet iS5 FTIR spectrometer (Fisher Scientific, Waltham, MA, USA) with a diamond crystal iD7 ATR accessory. Spectra were obtained as the average of 128 measurement cycles in the spectral range of 4000–550 cm^−1^ with a data interval of 0.964 cm^−1^. An atmospheric suppression function was employed to eliminate changes in ambient CO_2_ and H_2_O concentrations.

#### 3.2.6. X-Ray Diffraction (XRD) Technique

The XRD analysis was performed with an XRDynamic 500 diffractometer (Anton Paar, Graz, Austria) using a Cu lamp with Kα 1.54 Å wavelength excitation (40 kV, 50 mA) and a Si Pixos 2000 detector counting in 0D mode. A parallel beam collimator with a measurement radius of 360 mm was used. The measured diffractograms were analyzed with HighScore Plus software, version: 3.0e (3.0.5), PANalytical B. V., Almelo, the Netherlands, using PDF 4+ and COD23 databases.

#### 3.2.7. X-Ray Photoelectron Spectroscopy (XPS) Analysis of the Scaffolds

The elemental composition of the material surface was analyzed by X-ray photoelectron spectroscopy (XPS) using the ESCAProbeP spectrometer (Omicron Nanotechnology Ltd., London, United Kingdom). The atomic concentrations of the elements were determined from the individual peak areas using CasaXPS software (version 2.3.17.PR1.1., Casa Software Ltd., Teignmouth, United Kingdom). The spectra were measured with polymer analysis by XPS, i.e., with relatively low power of the X-ray source (75 W), using monochromatic X-ray radiation (1486.7 eV). The data were processed by the CasaXPS software, and the measured spectra were compared with our other results and reference analysis and with the NIST database. Peak fitting was based on the capabilities of the CasaXPS software.

#### 3.2.8. Mechanical Testing of Scaffolds

PLA/PCL foam specimens were mechanically tested. The foams were fabricated into experimental specimens of approximately cuboid shape (see Figure 1). The dimensions of the experimental specimens were as follows: width A 11 ± 2 mm, height B 12.1 ± 1.7 mm, thickness H 7.7 ± 2.4 mm, and area S (=A × B) of of 134 ± 35.5 mm^2^, to which a force was applied. Six types of foams were measured, and each type has 8 valid measurements (8 specimens). For each group of specimens, the values of the basic dimensions are given in Table 3. The foam specimens were mechanically loaded by compression in a Zwick/Roell biaxial test system designed for the mechanical testing of soft tissues and polymers. The specimen was compressed between two plates. An HBM U9C 50N force sensor was placed on each side, and the forces F1 and F2 were measured as shown in Figure 1. The resulting force was determined as the arithmetic mean of these two forces, F = (F1 + F2)/2. The loading and unloading rate was 0.1 mm/s (Figure 1). Markers were placed on the plates. The displacement of the markers was monitored by a video extensometer. From these displacements, the specimen deformations ε=(H−h)/H were calculated, where H is the initial and h is the deformed thickness of the specimens. The average strain rate was 0.0071/s. The sampling frequency of forces and displacements was 20 Hz. All specimens were loaded to a force limit of 20 N.

The engineering stress was calculated from the experimental data according to the relation σ=F/S, where F is the actual force and S is the reference (initial) area, S=A×B. The mechanical response was non-linear. This is related to the porosity of the specimens and the change in pore size during loading. The point of highest loading “end” was identified. The loading part of the experimental data was evaluated (Figure 1, right). The average strain of the specimens was 0.511 ± 0.081, and the stress was 163.8 ± 54.8 kPa.

**Scheme 1 ijms-26-02974-sch001:**
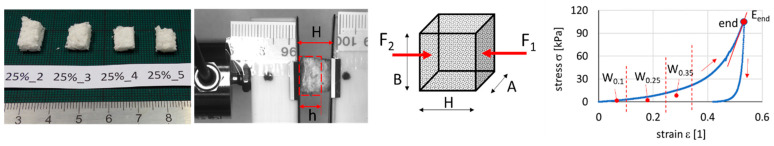
Experimental specimens and evaluation scheme. From left to right: example of experimental specimens of group 25%, specimen installation in the experimental system and marking the deformed configuration (dashed line), scheme of specimen dimensions and loading, and scheme of evaluation of mechanical properties, where W is the strain energy at selected strain states, E is the compressive modulus for the maximum value of the strain obtained, and “end” is the point of maximum loading of the specimen.

**Table 3 ijms-26-02974-t003:** Basic geometric dimensions of groups of experimental foam samples tested.

Sample/Parameter	A [mm]	B [mm]	H [mm]	S [mm^2^]
NM	13.42 ± 0.93	12.24 ± 0.53	9.24 ± 0.59	164.4 ± 15.7
0%	11.81 ± 0.54	12.75 ± 0.82	8.39 ± 0.72	150.5 ± 10.5
10%	11.84 ± 1.18	13.74 ± 0.93	10.01 ± 0.4	162.8 ± 21.4
25%	7.67 ± 0.48	10.14 ± 1.24	5.2 ± 0.39	77.8 ± 10.2
50%	9.92 ± 0.47	10.31 ± 0.7	4.02 ± 0.2	102.1 ± 6.6
100%	11.25 ± 1.21	13.15 ± 1.79	9.55 ± 0.63	146.6 ± 14.2

From the experimental stress–strain σ−ε characteristics, the average characteristics were calculated with the indicated confidence intervals (Figure 14). For given strain values, stress values were obtained by linear interpolation for all σ−ε characteristics. These could then be averaged and the confidence intervals calculated. The averaged characteristics represent the mechanical response of the specimens in a given group.

The highest strain achieved by all specimens tested was 35% (ε = 0.35). In the strain interval 0–0.35, the different groups can be compared. Strain values of 0 (initial), 0.1, 0.25, and 0.35 were selected, for which strain energy W values were calculated from the σ−ε characteristics (Figure 1). This quantity characterizes the amount of energy input required to deform the porous foams. The compressive elastic modulus E_end_ was also determined for the maximum strain, i.e., the maximum load achieved. At this point, it was assumed that the porosity of the material was already very small due to the loading. The compressive modulus of elasticity is close to the material parameter of a solid material with minimal air pores for a given deformation state.

The average characteristics and compressive modulus show the expected trend in the mechanical response of the foam groups investigated for higher strain values (Figure 14 and Figure 16D). For a better analysis of the mechanical properties at lower strains, where the σ−ε characteristics are close together, the strain energies were calculated and compared (Figure 16 A–C). To compare the mechanical response at the maximum applied load strain ε_end_ and stress σ_end_, see Figure 15.

For statistical analysis, the thicknesses of the specimens H, the area of the specimens S to which the compressive force F was applied, the values of the strain energy W for the selected deformation states, the values of the strain ε_end_, the stress σ_end_, and the compressive modulus E_end_ for the maximum load of the specimens were compared between the tested groups of foams. Statistical analysis was performed using the Real Statistics Resource Pack software (Release 8.9.1), Copyright (2013–2023) Charles Zaiontz, www.real-statistics.com, integrated in MS Excel. The normality of the data distribution of the compared groups was analyzed using the Shapiro–Wilk test and the d’Agostino–Pearson test. All data, except for the thickness of the 10% group, had normal distribution. ANOVA—single factor was chosen for statistical analysis. Tukey HSD/Kramer was applied as a post hoc test. All statistical tests were performed at the 95% significance level, i.e., *p* = 0.05. In the figures, the symbol “N” indicates statistical insignificance, i.e., the groups are similar.

### 3.3. Cellular Component, Isolation, and Characterization

#### 3.3.1. Adipose-Derived Stem Cells (ASCs)

ASCs were isolated from a lipoaspirate obtained from subcutaneous abdominal adipose tissue in accordance with the tenets of the Declaration of Helsinki on experiments involving human tissues and under the ethical approval of the Ethics Committee of the „Na Bulovce“ Hospital, Prague, Czech Republic. Written informed consent was obtained from the healthy donor (female, 41 years of age) prior to liposuction. A procedure previously published by Estes et al. [65] and modified in our studies [7,50] was used. Briefly, the lipoaspirate was washed several times with phosphate-buffered saline (PBS; Sigma-Aldrich, Merck Life Science Ltd., Prague, Czech Republic) in order to remove the contaminating blood cells from the adipose tissue. ASCs were released from the lipoaspirate by incubation for 1 h in a 0.1% (wt/vol) solution of collagenase type I (Worthington Biochemical Corp., Lakewood, NJ, USA) in PBS containing 1% (vol/vol) bovine serum albumin (BSA), at a lipoaspirate/collagenase solution ratio of 1:1. Incubation was performed for one hour at 37 °C with continuous mixing at 150 rpm. After digestion and centrifugation at 300 g for 5 min at 21 °C, the supernatant was removed, the cell suspension was transferred into a sterile tube, and 10 mL of incubation medium, namely Dulbecco’s modified Eagle’s Medium (DMEM) + 10% of fetal bovine serum (FBS; both from Gibco, Thermo Fisher Scientific, Waltham, MA, USA) + fibroblast growth factor-2 (FGF-2; 10 ng/mL; GenScript Biotech Co., Piscataway, NJ, USA, Cat. No. Z03116-1) + gentamicin (40 μg/mL, LEK, Ljubljana, Slovenia), was added. After mixing and centrifugation at 300 g for 5 min at room temperature (21 °C), the supernatant was removed, and the cell suspension was filtered through a 100 μm pore cell strainer (Biologix, Saint Louis, MO, USA) into a sterile vessel. In this way, mature adipocytes and remnants of digested tissue were removed, and the remaining stromal vascular fraction containing the ASCs was diluted with the above incubation medium and seeded into culture flasks (at a density of 0.16 mL of the original lipoaspirate per cm^2^).

ASCs in passage 2 were characterized by flow cytometry (Accuri C6 Flow Cytometer, BD Biosciences, Franklin Lakes, NJ, USA) to confirm the presence of typical mesenchymal stem cell markers (CD105, CD90, CD73, CD29) and the absence of markers of other cell types, such as hematopoietic cells and endothelial cells (CD45, CD34, CD31). Phycoerythrin-, FITC-, Alexa 488-, or Alexa 647-conjugated monoclonal antibodies against CD105, CD90, CD73, CD29, CD146, CD45, CD34, and CD31 were used. The percentage of cells positive for stem cell-specific CD markers was mostly higher than 98%. More details can be found in our previous studies [7].

#### 3.3.2. Human Umbilical Vein Endothelial Cells (HUVECs)

HUVECs were purchased from PromoCell (Heidelberg, Germany, Cat. No. C-12200-single donor) and expanded in endothelial cell growth medium 2 (EGM-2), which was prepared from endothelial cell basal medium 2 (EBM-2, PromoCell, Heidelberg, Germany, Cat. No. C-22111), supplemented with the growth medium 2 supplement pack (PromoCell, Heidelberg, Germany, Cat. No. C-39211) containing hydrocortisone, heparin, ascorbic acid, epidermal growth factor (EGF), vascular endothelial growth factor (VEGF), insulin-like growth factor-1 (IGF-1), FGF-2, 2% of FBS, and also with 1% antibiotic antimycotic solution (*v*/*v*, A5955, Sigma-Aldrich, Saint Louis, MO, USA).

### 3.4. Cell Cultivation

#### 3.4.1. Cultivation of ASCs

The sterile scaffolds were cut into 10 × 10 × 3 mm samples, inserted into 24-well polystyrene plates (Techno Plastic Products, Trasadingen, Switzerland), and seeded with ASCs at a density of 120,000 cells/sample. Cells on the samples were cultured in the above-mentioned DMEM medium, supplemented with 10% FBS (both from Gibco, Thermo Fisher Scientific, Waltham, MA, USA), FGF2 (10 ng/mL; GenScript Biotech Co., Piscataway, NJ, USA, Cat. No. Z03116-1), and gentamicin (40 μg/mL, LEK, Ljubljana, Slovenia) at 37 °C in a humidified air atmosphere with 5% CO_2_. Four samples were tested in two independent experiments for each scaffold type. The culture medium was changed every 3 days.

Cells were cultured under both static and dynamic conditions. After 6 days of static cultivation, when the cells were sufficiently spread on the scaffolds, the plates containing the cell–material constructs were either further cultured statically for a further 4 days or transferred to the Stuart Mini Orbital Shaker SSM1 (Cole Parmer, Vernon Hills, Il, USA). This device generated fluid shear stress by circular motion of a horizontal platform with an orbit of 16 mm and a speed of 60 revolutions per minute (rpm), i.e., 1 Hz. These dynamic conditions were expected to improve the penetration of cells into the scaffolds, their proliferation, the supply of nutrients and oxygen, and the removal of waste products. In our earlier studies, similar conditions increased the metabolic activity of human osteoblast-like MG-63 cells on nanofibrous scaffolds [14] and osteogenic differentiation of human bone marrow mesenchymal stem cells in cultures on Ti6Al4V alloy coated with zeolite (silicalite-1) films [58].

The osteogenic differentiation of ASCs was also promoted biochemically, i.e., by an osteogenic culture medium. First, the cells were pre-cultured on the scaffolds in the growth medium described above, i.e., DMEM + 10% FBS + FGF-2 (10 ng/mL) + gentamicin (40 μg/mL) for 2 days and then exposed to the osteogenic medium consisting of DMEM + 10% FBS and gentamicin, further supplemented with dexamethasone (10 nM), beta-glycerol phosphate (10 mM) and ascorbic acid (50 μg/mL), which has been shown to be efficient for osteogenic differentiation of ASCs in our previous studies [7,50].

#### 3.4.2. Co-Cultivation of ASCs and HUVECs

To develop the pre-vascularization of the scaffolds, ASCs were co-cultured with HUVECs. ASCs and HUVECs were seeded together on the samples at a total number of 240,000 cells per sample, but in different ratios, i.e., 1:1 (120,000 cells of each cell type), 1:2 (80,000 ASCs and 160,000 HUVECs), 1:5 (40,000 ASCs and 200 HUVECs), and 2:1 (160,000 ASCs and 80,000 HUVECs). Both cell types were co-cultured in an EGM-2 medium with the above-mentioned supplements, i.e., hydrocortisone, heparin, ascorbic acid, EGF, VEGF, IGF-1, FGF-2, 2% of FBS, and 1% antibiotic antimycotic solution (PromoCell, Heidelberg, Germany). In some scaffolds, cell penetration into the interior of the scaffold was facilitated by centrifuging the cells with the scaffolds during seeding (50 g, 3 min). The pre-vascularization of the scaffolds was further enhanced by dynamic cultivation. After 2 days of pre-cultivation under static conditions to allow adhesion spreading and initial growth of the cells on the scaffolds, the well plates containing the samples were transferred to a Stuart Mini Orbital Shaker SSM1 (orbit 16 mm 60 rpm) for an additional 4 days.

### 3.5. Cell Visualization and Confocal Microscopy

Cells on the scaffolds were visualized by fluorescence staining of the filamentous actin (F-actin), to assess the assembly of the actin cytoskeleton in the cells, the shape and spreading of the cells on the tested materials, and the colonization of the inner space of the scaffold. Prior to staining, the cells on the samples were rinsed with PBS and fixed with 4% paraformaldehyde in PBS for 30 min at room temperature. The cells were then permeabilized with 0.1% Triton X-100 and 1% BSA in PBS for 20 min, followed by incubation in 1% Tween-20 in PBS for 20 min. Finally, the cells were stained with Phalloidin conjugated with Alexa Fluor 488 (green fluorescence, Cat. No. A12379) or Alexa Fluor 568 (red fluorescence, Cat. No. A12380; both from Thermo Fisher Scientific, Waltham, MA, USA; 1000× diluted in PBS). Cell nuclei were counterstained with 4′,6-diamidin-2-fenylindol (DAPI; Sigma-Aldrich, Cat. No. 32670, 20 μg/mL) added to the phalloidin solution. The samples were stained for 1 h at room temperature.

Selective visualization of HUVECs was performed by incubating live cells with CellTrackerTM Green CMFDA (Thermo Fisher Scientific, Waltham, MA, USA, Cat. No. C7025) added to the culture medium at a concentration of 25 µM for 45 min.

Micrographs of the cells on the materials were taken using an Andor Dragonfly 503 scanning disk confocal microscope, equipped with a Zyla 4.2 PLUS sCMOS camera (Andor Technology Ltd., Belfast, UK) and an HC PL APO 20×/0.75 IMM CORR CS2 objective, allowing 3D visualization of the constructs.

### 3.6. Cell Number Counting and Statistics

From each sample group, 5–6 images of cell nuclei were obtained. The number of cell nuclei, which was considered the number of cells, was manually counted from the Maximum Image Projection (MIP) using Fiji ImageJ software (National Institutes of Health, Bethesda, MD, USA; version FIJI 2.0.0-rc-68/1.52 h). The number of cells was presented as the Mean ± Standard Error of the Mean (S.E.M.) of 3–6 images. Statistical significance of differences in cell number between samples was assessed using One Way ANOVA and the Student–Newman–Keuls method. A value of *p* ≤ 0.05 was considered significant. Statistical analysis was performed using SigmaPlot 14.0 software (Systat Software Inc., San José, CA, USA).

## 4. Conclusions and Further Perspectives

In this study, soft three-dimensional foam-like scaffolds were prepared by freeze-drying from solutions of PLA and PCL blends (ratio 3/5) in 1,4-dioxane with or without the addition of sodium chloride porogen. The freeze-dried foams exhibited interconnected porous aerogel properties. The hierarchical structure with bimodal micro–macro pore size distribution was obtained by adding the porogen with defined crystal size distributions to the solutions prior to freeze-drying and leaching the porogen crystals from the freeze-dried foams with demineralized water. The addition of the second porogen, Klucel™ E hydroxypropyl cellulose, at concentrations of 10, 25, 50, and 100% *w*/*w* relative to the total concentration of PLA/PCL, further enlarged the pores and prevented possible shrinkage. SEM revealed that the use of the NaCl porogen alone resulted in scaffolds with a regular internal structure of parallel-oriented pores, whereas the addition of Klucel to NaCl to further enlarge the pores completely disrupted this regular pore organization. Micro-CT analysis (parameter structure separation) showed that the size of the macropores in scaffolds prepared with Klucel was, on average, larger (483–539 µm) than without Klucel (298 µm), although all scaffolds also contained pores smaller than 200 µm, mainly around 40–100 µm in diameter. Surprisingly, the scaffolds prepared with Klucel had a slightly lower porosity (approximately 89–92%) than the Klucel-free scaffolds (93%), but these differences were not significant. All scaffolds had an open porosity of approximately 90%, and the pores were interconnected. BET analysis revealed that in addition to macro- and micropores, the scaffolds also contained nanopores (mainly wih a half-pore size of 12–30 Å, i.e., 2.4–6 nm in diameter). This nanoporosity was higher in the scaffolds prepared with Klucel and increased the water uptake of the scaffolds, i.e., their hydrophilicity. This hydrophilicity was probably also enhanced by incomplete leaching and some retention of Klucel in the scaffolds, as suggested by FTIR and further confirmed by XRD and XPS analyses. This retention, together with the disruption of regular pore organization, probably hindered cell penetration into the scaffolds. In a conventional static cell culture system, the number of adipose-derived stem cells (ASCs) on Klucel-treated scaffolds was similar (day 6) or even lower (day 10) than on Klucel-free scaffolds. However, in a dynamic cell culture system with fluid shear stress and perfusion of the scaffolds with culture medium, the number of cells and their penetration into the scaffolds increased significantly in scaffolds prepared with 25% Klucel. Dynamic cultivation also enhanced the formation of pre-vascular structures in the scaffolds when ASCs were co-cultured with endothelial cells.

The attractiveness of the scaffolds for cell colonization was also enhanced by their mineralization in simulated body fluid, which was evident in both standard and osteogenic cell culture media. Moreover, the addition of Klucel allowed us to tailor the mechanical properties of the scaffolds for soft- to hard-tissue engineering applications. Our newly prepared foam-like scaffolds are therefore promising for the engineering of a variety of pre-vascularized tissues for reconstructive surgery, such as adipose tissue (less stiff and more pliable scaffolds prepared without Klucel or with 10% Klucel) or bone tissue (stiffer and mechanically more resistant scaffolds prepared with 25% or 50% Klucel).

## Data Availability

The data presented in this study are available upon request from the corresponding authors and in the Zenodo repository at https://doi.org/10.5281/zenodo.15051471.

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
