# Peer review of "Composite Polylactide/Polycaprolactone Foams with Hierarchical Porous Structure for Pre-Vascularized Tissue Engineering"

_ijms, 2025, doi:10.3390/ijms26072974_

Round 1
Reviewer 1 Report
Comments and Suggestions for Authors
This manuscript provides a good preparation of scaffold for tissue regeneration. A few minor corrections will make your manuscript more complete. The following comments and suggestions should be helpful for the improvement of the paper.
1. It is misleading to describe PLA as hydrophilic and PCL as crystalline in Introduction part.
2. In Figure 10, I recommend providing the peaks of NaCl and Klucel to explain them together. Also, the description of the peaks in the text is insufficient.
3. Which group has the most ideal mechanical properties, and why does the author think so?
4. In Figure 18, cell viability has decreased. I wonder if using Klucel is really the right thing to do for cell regeneration.
Author Response
- It is misleading to describe PLA as hydrophilic and PCL as crystalline in Introduction part.
Reply:
The Reviewer is probably referring to the sentences „PLA is more hydrophilic, making it more suitable for cell adhesion and growth [31], but it has a glass transition temperature above room temperature, making it hard and brittle. PCL, however, is crystalline and has a low glass transition temperature, making it tough but with a modulus an order of magnitude lower than PLA [32]“ on lines 146-150.
In the field of physical chemistry, it is generally accepted that if the water contact angle is less than 90°, the solid surface is considered hydrophilic, and if the water contact angle is greater than 90°, the solid surface is considered hydrophobic (Ahmad et al. 2018). From this point of view, PLA still falls into the category of hydrophilic polymers because, according to the literature, its water contact angle is in the range of 60-85° (Tümer et al. 2022, Orue et al. 2016; for a review, see Galindo and Urena-Nunez 2018). In a study by Chen et al. (2022), PLA is also classified as "hydrophilic in nature" with a contact angle of 75°. At the same time, we recognize that the terms hydrophilic and hydrophobic are relative. For example, compared to PEO, PLA will be hydrophobic. In this article, we just wanted to express that PLA can be considered more hydrophilic than PCL, which has a water contact angle of 105° (Torres et al. 2020=reference 32 in the manuscript). In the aforementioned study, blending PCL with PLA reduced this water drop contact angle to 87°, a more acceptable value for cell adhesion and growth. It is known that cells adhere, migrate and proliferate optimally on moderately wettable surfaces with a contact angle of approximately 40-70° (Bacakova et al. 2011, Torres et al. 2020). Therefore, in our present study, we also used a blend of PCL with PLA to fabricate polymeric scaffolds for tissue engineering purposes.
The sentence “PCL, however, is crystalline and has a low glass transition temperature, making it tough but with a modulus an order of magnitude lower than PLA” was adopted from the study by Broz et al. 2003 (reference 33 in the manuscript). However, according to other studies, both PCL and PLA are defined as semi-crystalline (Sun et al. 2014, Fortelny et al. 2019), although the crystallinity of PCL is reported to be higher (about 35%) than that of biodegradable PLA, widely used for tissue engineering purposes, which is about 10% (Zhang et al. 2012). Nevertheless, PLA can also be prepared with high crystallinity, e.g. by using a nucleating agent (Fortelny et al. 2019) or high pressure (Zhang et al. 2012). Therefore, we decided to omit the word "crystalline", which may be confusing and is not relevant to the focus of our study. The final wording of the sentences in the revised manuscript is therefore the following:
“PLA is more hydrophilic than PCL, making it more suitable for cell adhesion and growth [33], but it has a glass transition temperature above room temperature, making it hard and brittle. PCL, however, has a low glass transition temperature, making it tough but with a modulus an order of magnitude lower than PLA [33]“ (Page 4)
Ahmad, D., van den Boogaert, I., Miller, J., Presswell, R., & Jouhara, H. (2018). Hydrophilic and hydrophobic materials and their applications. Energy Sources, Part A: Recovery, Utilization, and Environmental Effects, 40(22), 2686–2725. https://doi.org/10.1080/15567036.2018.1511642
Bacakova L, Filova E, Parizek M, Ruml T, Svorcik V. Modulation of cell adhesion, proliferation and differentiation on materials designed for body implants. Biotechnol Adv. 2011;29(6):739-67. doi: 10.1016/j.biotechadv.2011.06.004.
Chen Y, Tang T, Ayranci C. Moisture-induced anti-plasticization of polylactic acid: Experiments and modeling. J Appl Polym Sci. 2022;e52369. doi:10.1002/app.52369
Galindo S, Ureña-Nuñez F: Enhanced surface hydrophobicity of poly(lactic acid) by Co60 gamma ray irradiation. Revista Mexicana de Física 64 (2018) 1–7
Orue A, Eceiza A, Peña-Rodriguez C Arbelaiz A. Water Uptake Behavior and Young Modulus Prediction of Composites Based on Treated Sisal Fibers and Poly(Lactic Acid). Materials 2016, 9, 400; doi:10.3390/ma9050400
Tümer EH, Erbil HY, Akdoǧan N. Wetting of Superhydrophobic Polylactic Acid Micropillared Patterns. Langmuir 2022, 38, 10052−10064, doi: 10.1021/acs.langmuir.2c01708
Sun H, Yu B, Han J, Kong J, Meng L, Zhu F. Microstructure, Thermal Properties and Rheological Behavior of PLA/PCL Blends for Melt-blown Nonwovens. Polymer Korea 2014; 38(4):477-483; doi: 10.7317/pk.2014.38.4.477
Fortelny I, Ujcic A, Fambri L and Slouf M (2019) Phase Structure, Compatibility, and Toughness of PLA/PCL Blends: A Review. Front. Mater. 6:206. doi: 10.3389/fmats.2019.00206
Zhang J, Yan D-X, Xu J-Z, Huang H-D, Lei J, Li Z-M. Highly crystallized poly (lactic acid) under high pressure. AIP Advances 2012; 2 (4): 042159. doi: 10.1063/1.4769351
- In Figure 10, I recommend providing the peaks of NaCl and Klucel to explain them together. Also, the description of the peaks in the text is insufficient.
Reply: The FTIR spectra of the materials with and without NaCl porogen were very similar and virtually identical, indicating that the NaCl porogen was completely washed out of the material, leaving no residual spectrum. Even the application of Klucel, i.e. an additional porogen, did not cause any significant changes in the FTIR spectra of our materials. However, we observed a slight increase in the intensity ratio of the PLA bands, i.e. the PLA vibrations C=O at 1750 cm-1, C-H at 1460 cm-1 and C-O-C at 1090 cm-1, as described in the manuscript. These changes were similar to those observed after biomimetic mineralization of the scaffolds in the simulated body fluid and were explained by a further increase in the wettability of PLA by the application of mineral components and hydrophilic Klucel. From this point of view, we could not exclude the possibility of retention of at least a small amount of Klucel in the scaffolds. Therefore, we performed XRD and XPS as requested by Reviewer 2. Both analyses showed a complete washout of NaCl, but XRD indicated some retention of Klucel in the scaffolds (see the response to the last comment of Reviewer 2 and new Figures 10 and 11 in the manuscript).
In addition, a new Figure 9 with better-described peaks has been added to the manuscript, and this figure has also been better described in the text (pages 14 and 15).
- Which group has the most ideal mechanical properties, and why does the author think so?
Reply: The most advantageous mechanical properties depend on the intended use of the scaffold. Less stiff and more pliable scaffolds prepared without Klucel or with 10% Klucel could be advantageous for soft tissue engineering. It is known that soft tissues have elastic moduli E (mostly tested in tension) ranging from 0.1 MPa to 1 MPa (Liu et al. 2015). Related to this are the strain energy values, which are also many times lower. Our scaffolds, especially those prepared without Klucel (both mineralized and non-mineralized), reach the soft tissue modulus limit, about 1 MPa.
Among the soft tissues that could be reconstructed with these foam scaffolds, adipose tissue, which can be used for breast reconstruction after tumor removal, occupies an important place. Current clinical practice for breast reconstruction is to use acellular silicone implants or to fill defects with autologous fat grafts. Both approaches have disadvantages - silicone implants are associated with the risk of local and systemic immune reactions, i.e. breast implant illness (BII, for a review, see Tervaert et al. 2024), while fat grafts are often at least partially resorbed, significantly reducing their beneficial effect (Molitor et al. 2021). These problems could be overcome by using resorbable scaffolds seeded with autologous cells and pre-vascularized. These scaffolds are then progressively replaced with the patient's new regenerated tissue to permanently fill the defect and remove the artificial material.
A similar effect could be observed in bone tissue engineering with a pre-vascularized and autologous cell-populated polymeric scaffold. The stiffer and mechanically more resistant scaffolds prepared with 25% or 50% Klucel may be suitable for this purpose. However, it should be noted that the Young's modulus (E) of bone tissue is in the range of units to tens of GPa, whereas our scaffolds prepared with 25% or 50% Klucel have an E of only about 2.5 to 3 MPa. Thus, they may be more suitable for filling bone defects (some improvement in mechanical properties can be expected after bone tissue has formed inside the scaffolds) than for load-bearing applications, such as bone-integrating parts of large joint replacements. These applications still require metallic materials, such as titanium and its alloys. Virtually the only polymer that could be used in place of metals in high-stress applications is PEEK, especially when reinforced with carbon fibers (for a review, see Arevalo et al. 2023).
The question of which group has the most ideal mechanical properties is partially answered at the end of Chapter 2.8. “Mechanical testing of scaffolds” (pages 19-20), and also in Chapter 4 “Conclusions and further perspectives” (pages 36-37), and this answer has been further enriched by some of the aspects mentioned in this response to the Reviewer.
Liu J, Zheng H, Poh PS, Machens HG, Schilling AF. Hydrogels for Engineering of Perfusable Vascular Networks Int J Mol Sci. 2015;16(7):15997-6016. doi: 10.3390/ijms160715997.
Tervaert JWC, Shoenfeld Y, Cruciani C, Scarpa C, Bassetto F. Breast implant illness: Is it causally related to breast implants? Autoimmun Rev. 2024;23(1):103448. doi: 10.1016/j.autrev.2023.103448.
Molitor M, Trávníčková M, Měšťák O, Christodoulou P, Sedlář A, Bačáková L, Lucchina S. The Influence of High and Low Negative Pressure Liposuction and Various Harvesting Techniques on the Viability and Function of Harvested Cells-a Systematic Review of Animal and Human Studies. Aesthetic Plast Surg. 2021;45(5):2379-2394. doi: 10.1007/s00266-021-02249-9.
Arevalo S, Arthurs C, Molina MIE, Pruitt L, Roy A. An overview of the tribological and mechanical properties of PEEK and CFR-PEEK for use in total joint replacements. J Mech Behav Biomed Mater. 2023;145:105974. doi: 10.1016/j.jmbbm.2023.105974.
- In Figure 18, cell viability has decreased. I wonder if using Klucel is really the right thing to do for cell regeneration.
Reply: Figure 18 (now 20) does not show cell viability, but cell population density, i.e. the number of cells per cm2 calculated from the Maximal Imaging Projection (MIP) images of the tested materials. However, the reviewer is correct that this cell count is lower in Klucel-treated scaffolds than in Klucel-untreated scaffolds. Although Klucel increased the pore diameter (from 298 µm in Klucel-free scaffolds to 483 - 539 µm in Klucel-treated scaffolds), it significantly deformed the originally regular and axially oriented internal porous structure of the scaffolds (see SEM images in Figures 1-3). In addition, as mentioned in the comments to the FTIR images, it could not be excluded that some Klucel, at least in small amounts, was retained inside the pores. The retention of Klucel was then confirmed by XRD analysis. Therefore, both the deformation of the porous structure and the retention of Klucel could complicate the penetration of cells into the Klucel-treated scaffolds.
However, the use of a dynamic cell culture system significantly improved cell penetration into the scaffolds so that by day 10 after seeding, the significant difference in cell numbers on Klucel-free and Klucel-treated scaffolds disappeared, at least for scaffolds prepared with 25% Klucel (Figure 20). In addition, cells on Klucel-treated scaffolds were generally better spread and more homogeneously distributed, especially on scaffolds prepared with medium and higher concentrations of Klucel, i.e., 25% and 50% (Figure 19, 21). This may be due, at least in part, to the increased hydrophilicity of the Klucel-treated scaffolds as suggested by FTIR. Moderate hydrophilicity of the material surface is known to promote the adsorption of cell adhesion-mediating molecules (such as vitronectin and fibronectin, present in the serum supplement of the culture medium) in a bioactive physiological conformation, which promotes the binding of these molecules by cell adhesion receptors. These receptors recognize specific amino acid sequences in these molecules, such as Arg-Gly-Asp (RGD), which then leads to improved cell adhesion, spreading, migration, and subsequent proliferation of cells on the scaffolds (for a review, see Bacakova et al. 2011=reference 43 in this manuscript; Musilkova et al. 2015; see also the response to Reviewer 2's comment 1).
Last but not least, the Klucel-treated scaffolds, at least those prepared with 25% and 50% of Klucel, also improved their mechanical strength (Figures 14-16) and were able to support the development of pre-vascular structures when seeded with adipose-derived mesenchymal stem cells (ASCs) along with endothelial cells (Figure 24).
Thus, although the preparation of PLA/PCL scaffolds with Klucel had some disadvantages, such as partial blockage and deformation of the pores, more difficult cell penetration into the scaffolds and the need for dynamic cell cultivation, it allowed the preparation of different scaffolds for specific applications. Klucel-free scaffolds are more suitable for soft tissue engineering, while Klucel-treated scaffolds (25 and 50%) are more suitable for hard tissue engineering. In addition, Klucel reduces the hydrophobicity of the material surface and improves cell adhesion and spreading.
Some of these remarks have been added to the Results and Discussion section of the manuscript, namely on pages 23, 24 and 25.
Musilkova J, Kotelnikov I, Novotna K, Pop-Georgievski O, Rypacek F, Bacakova L, Proks V. Cell adhesion and growth enabled by biomimetic oligopeptide modification of a polydopamine-poly(ethylene oxide) protein repulsive surface. J Mater Sci Mater Med. 2015;26(11):253. doi: 10.1007/s10856-015-5583-3.

Reviewer 2 Report
Comments and Suggestions for Authors
1、“Nanoscale surface roughness, i.e.irregularities equal to or less than 100 nm, is also beneficial because it mimics the nanoarchitecture of the native ECM and promotes the adsorption of cell adhesion mediating proteins, such as vitronectin and fibronectin, in a suitable geometric conformation accessible to cell adhesion receptors.” on page 7, from line 234 to line 237. The current statement lacks a comprehensive explanation regarding the underlying mechanisms that facilitate favorable integration between the implant and surrounding tissues. It is recommended that supporting evidence be provided by citing relevant literature or conducting supplementary experimental studies to validate the findings.
2、“Thus, the pore size of our scaffolds, especially those prepared with Klucel, which averaged from 483 μm to 539 μm, can be considered suitable for bone tissue engineering. In our earlier study and studies by other authors, pores of the size 400-600 μm appeared to be optimal for this purpose.”on page 8, from line 299 to line 301.What is the scientific rationale for the 483 ~ 539 μm pore size being particularly suitable for bone tissue engineering applications?
3、The manuscript mentions that the macroporosity of the scaffold increases cell ingrowth, why does it increase, and which of the associated proteins rises in expression?
4、It is suggested that additional experimental data from XRD and XPS be added to further explain the composition of the scaffolds.
Author Response
1、“Nanoscale surface roughness, i.e.irregularities equal to or less than 100 nm, is also beneficial because it mimics the nanoarchitecture of the native ECM and promotes the adsorption of cell adhesion mediating proteins, such as vitronectin and fibronectin, in a suitable geometric conformation accessible to cell adhesion receptors.” on page 7, from line 234 to line 237. The current statement lacks a comprehensive explanation regarding the underlying mechanisms that facilitate favorable integration between the implant and surrounding tissues. It is recommended that supporting evidence be provided by citing relevant literature or conducting supplementary experimental studies to validate the findings.
Reply: Materials with nanoscale surface roughness are known to promote cell adhesion, spreading, and subsequent growth and differentiation. One of the first to describe this phenomenon was Webster and coworkers, who observed increased adhesion of osteoblasts to nanophase ceramics (Webster et al. 2000a, b). The reason is that the nanostructure of a material resembles the nanoarchitecture of the natural extracellular matrix (ECM), e.g., its organization into nanofibers, nanocrystals, nanoscale folds of ECM molecules, etc. On nanostructured surfaces, the ECM molecules that mediate cell adhesion are then adsorbed in an appropriate, nearly physiological geometric conformation that provides cell adhesion receptors with access to specific sites on the ECM molecules, such as amino acid sequences like Arg-Gly-Asp (RGD), that serve as ligands for these receptors. From this point of view, surface nanoroughness can be considered to act synergetically with the moderate hydrophilicity of the material surface, which also promotes the adsorption of cell adhesion-mediating molecules in bioactive physiological conformations.
Cell adhesion molecules include fibronectin, vitronectin, collagen and laminin. Under in vivo conditions, they are spontaneously adsorbed onto an implanted biomaterial from body fluids, e.g. interstitial fluid, while in in vitro cell culture systems, the main source of these molecules is the serum supplement of the cell culture medium. Interestingly, it is believed that materials with nanoscale surface roughness preferentially adsorb vitronectin due to its relatively small and linear molecule, making these surfaces particularly advantageous for bone tissue engineering. This is because vitronectin is preferentially bound by heparan sulfate proteoglycan molecules on the membrane of osteoblasts that recognize specific amino acid sequences, e.g. Lys-Arg-Ser-Arg (KRSR), in the heparin-binding domain of vitronectin.
This issue has been elaborated and schematically documented in detail in our previous studies, in particular in the review article by Bačáková et al. 2011 (reference 43 in this manuscript), a primary article by Musílková et al. 2015, and the book chapter by Bačáková et al. 2016. A more detailed explanation of this issue, together with relevant references, has been added to this manuscript (page 7).
Webster TJ, Ergun C, Doremus RH, Siegel RW, Bizios R. Specific proteins mediate enhanced osteoblast adhesion on nanophase ceramics. J Biomed Mater Res. 2000a Sep 5;51(3):475-83. doi: 10.1002/1097-4636(20000905)51:3<475::aid-jbm23>3.0.co;2-9.
Webster TJ, Ergun C, Doremus RH, Siegel RW, Bizios R. Enhanced functions of osteoblasts on nanophase ceramics. Biomaterials. 2000b Sep;21(17):1803-10. doi: 10.1016/s0142-9612(00)00075-2.
Bacakova L, Filova E, Liskova J, Kopova I, Vandrovcova M, Havlikova J. Nanostructured materials as substrates for the adhesion, growth, and osteogenic differentiation of bone cells. In: Nanobiomaterials in Hard Tissue Engineering. Applications of Nanobiomaterials. Volume 4, Ed. A. M. Grumezescu, Elsevier Inc., William Andrew Publishing, Oxford, Cambridge; Chapter 4, pp. 103-153, ISBN 978-0-323-42862-0; 2016
Musilkova J, Kotelnikov I, Novotna K, Pop-Georgievski O, Rypacek F, Bacakova L, Proks V. Cell adhesion and growth enabled by biomimetic oligopeptide modification of a polydopamine-poly(ethylene oxide) protein repulsive surface. J Mater Sci Mater Med. 2015;26(11):253. doi: 10.1007/s10856-015-5583-3.
2、“Thus, the pore size of our scaffolds, especially those prepared with Klucel, which averaged from 483 μm to 539 μm, can be considered suitable for bone tissue engineering. In our earlier study and studies by other authors, pores of the size 400-600 μm appeared to be optimal for this purpose.”on page 8, from line 299 to line 301.What is the scientific rationale for the 483 ~ 539 μm pore size being particularly suitable for bone tissue engineering applications?
Reply: It is well known that scaffolds for bone tissue engineering need to have larger pore diameters than those used for soft tissue or even cartilage engineering. This is because osteoblasts are often arranged in more complex systems than soft tissue cells. For example, the Haversian system of concentric lamellae of osteoblasts in cortical bone reaches diameters of 100-200 μm, so this size is considered minimal for pores in scaffolds designed for bone tissue engineering (for a review, see Karageorgiou and Kaplan 2005=reference 47 in the manuscript). However, our previous studies performed on PLGA scaffolds (Pamula et al. 2008, 2009; references 12 and 13 in the manuscript) and similar studies by other authors have shown that even a larger pore diameter, i.e., in the range of 400-600 μm, is optimal for bone tissue engineering. The reason is that in addition to the system of concentric lamellae, we must also consider the presence of mineralized bone matrix and vascular supply. The diameter of the pores in the scaffolds prepared with 25% and 50% of Klucel, namely 483 μm to 539 μm, is close to the mentioned optimal range and therefore we believe that these scaffolds would be suitable for bone tissue engineering. Fortunately, a larger pore size was also associated with higher mechanical strength in these scaffolds, while a smaller pore diameter in Klucel-free scaffolds was associated with higher pliability, so these scaffolds seem to be applicable in soft tissue engineering.
The explanation of the suitability of the pore size from 483 μm to 539 μm for bone tissue engineering has been added to the manuscript (pages 8-9).
3、The manuscript mentions that the macroporosity of the scaffold increases cell ingrowth, why does it increase, and which of the associated proteins rises in expression?
Reply: The macroporosity of a scaffold is defined as the presence of pores of tens and hundreds of micrometers in diameter, while the pores less than 10 µm are defined as micropores and less than 100 nm as nanopores (Bruzauskaite et al. 2016=reference 23 in the manuscript; Chanes-Cuevas et al. 2018).
Since the cells seeded on scaffolds in the form of their suspension in the medium are usually around 20 µm in diameter, it is clear that these cells are only able to migrate into macroporous scaffolds and only into those with pores with a diameter of higher tens of micrometers, or hundred(s)of micrometers. In our earlier study performed on PLGA scaffolds, pores 40 µm in diameter were „capped“ by human osteoblast-like MG 63 cells spreading on the surface of the scaffold without these cells penetrating into the interior of the scaffolds (Pamula et al. 2008=reference 12 in the manuscript).
As explained above, increasing the macropore diameter initially promotes cell migration into the interior of the scaffold and cell proliferation. Of course, this is accompanied by the synthesis of a number of specific proteins by these cells, such as extracellular matrix proteins like collagen. In the case of osteogenic cells, there is also the synthesis of other proteins typical of osteogenic cell differentiation, such as osteocalcin or the enzyme alkaline phosphatase, which is involved in bone matrix mineralization.
However, it must be acknowledged that macroporosity has a positive effect on cell ingrowth into the scaffold only up to a certain optimal limit. Too large a macropore diameter may be counterproductive. It can result in reduced mechanical stability and internal surface area of the scaffolds, limited cell adhesion, insufficient cell-to-cell contacts, reduced cell proliferation activity and mineralization, and consequently limited bone formation. However, the definition of pores that are too large for bone formation varies between materials - for example, in 3D printed ceramic materials, such pores were over 1.5 mm in diameter, whereas in collagen-based scaffolds, such pores were only 500 μm in diameter (Ghayor and Weber 2018, Yamahara et al. 2022).
In the manuscript (Abstract), however, the word "increased" has been replaced by "promoted" - adequate macroporosity not only increases cell ingrowth, but is a direct requirement for sufficient cell ingrowth into the scaffold - in other words, macroporosity enables and supports cell ingrowth.
Chanes-Cuevas OA, Perez-Soria A, Cruz-Maya I, Guarino V, Alvarez-Perez MA. Macro-, micro- and mesoporous materials for tissue engineering applications[J]. AIMS Materials Science, 2018, 5(6): 1124-1140. doi: 10.3934/matersci.2018.6.1124
Ghayor C, Weber FE. Osteoconductive Microarchitecture of Bone Substitutes for Bone Regeneration Revisited. Front Physiol. 2018;9:960. doi: 10.3389/fphys.2018.00960.
Yamahara S, Montenegro Raudales JL, Akiyama Y, Ito M, Chimedtseren I, Arai Y, Wakita T, Hiratsuka T, Miyazawa K, Goto S, Honda M. Appropriate pore size for bone formation potential of porous collagen type I-based recombinant peptide. Regen Ther. 2022;21:294-306. doi: 10.1016/j.reth.2022.08.001.
4、It is suggested that additional experimental data from XRD and XPS be added to further explain the composition of the scaffolds.
Reply: The XRD and XPS analyses have been performed and new Figures 11 and 12 have been added to the manuscript with corresponding text in the Results and Discussion (pages 15, 16) and Materials and Methods (page 32) sections. XRD showed that the NaCl porogen was completely washed out of the scaffolds, but XPS revealed a trace amount of Cl in the scaffolds prepared with 50% Klucel. Both XRD and XPS analyses also confirmed that some Klucel remained trapped in the scaffolds, as suggested by FTIR. This retention of Klucel was probably the reason for the lower cell colonization of the Klucel-treated scaffolds in the static culture system, which was only improved by dynamic cultivation (Figure 20). However, the use of Klucel in the preparation of the scaffolds also had its advantages. Klucel, i.e. highly hydrophilic hydroxyprolyl cellulose, modulated the wettability of the scaffolds to values suitable for cell adhesion and spreading, and importantly, it allowed us to modulate the mechanical properties of the scaffolds for specific applications. Less stiff and more pliable scaffolds prepared without Klucel or with a low concentration of Klucel (10%) could be used in soft tissue engineering, such as adipose tissue, while stiffer and mechanically more resistant scaffolds prepared with 25% or 50% Klucel could be applicable in bone and cartilage tissue engineering (see also the responses to the comments of Reviewer 1).

Round 2
Reviewer 2 Report
Comments and Suggestions for Authors
The comments made have been refined and the manuscript is original and innovative to a high degree and is recommended for publication.
Author Response
We thank the reviewer for his favorable opinion. We believe that the manuscript is now ready for publication in IJMS.